

# Saharan dust events in the European Alps: role on snowmelt and geochemical characterization

Biagio Di Mauro[1], Roberto Garzonio[1], Micol Rossini[1], Gianluca Filippa[2], Paolo Pogliotti[2], Marta Galvagno[2], Umberto Morra di Cella[2], Mirco Migliavacca[3], Giovanni Baccolo[1,4], Massimiliano Clemenza[4,5], Barbara Delmonte[1], Valter Maggi[1], Marie Dumont[6], François Tuzet[6,7], Matthieu Lafaysse[6], Samuel Morin[6], Edoardo Cremonese[2], and Roberto Colombo[1]

[1]Earth and Environmental Sciences Department, University of Milano-Bicocca, 20126 Milan, Italy
[2]Environmental Protection Agency of Aosta Valley, Aosta, Italy
[3]Max Planck Institute for Biogeochemistry, Jena, Germany
[4]National Institute of Nuclear Physics (INFN), Section of Milano-Bicocca, Milan, Italy
[5]Department of Physics "G. Occhialini", University of Milano-Bicocca, 20126 Milan, Italy
[6]Univ. Grenoble Alpes, Université de Toulouse, Météo-France, CNRS, CNRM, Centre d'Etudes de la Neige, Grenoble, France
[7]UGA/CNRS, Institut des Géosciences de l'Environnement (IGE), Saint Martin d'Heres, France

*Correspondence to*: Biagio Di Mauro Ph.D. (biagio.dimauro@unimib.it)

**Abstract.** The input of mineral dust from arid regions impacts snow optical properties. The induced albedo reduction generally alters the melting dynamics of the snowpack, resulting in earlier snowmelt. In this paper, we evaluate the impact of dust depositions on the melting dynamics of snowpack in a high-altitude site (2160 m) in the European Alps (Torgnon, Aosta Valley, Italy) during three hydrological years (2013-2016). These years were characterized by several Saharan dust events that deposited significant amounts of mineral dust in the European Alps. We quantify the shortening of snow season due to dust deposition, by comparing observed snow depths and those simulated with the Crocus model accounting or not for the impact of impurities. The model was ran and tested using meteorological data from an Automated Weather Station. We propose the use of repeated digital images for tracking dust deposition and resurfacing in the snowpack. The good agreement between model prediction and digital images allowed us to propose the use of an RGB index (i.e. snow darkening index, SDI) for monitoring dust on snow using images from a digital camera. We also present a geochemical characterization of dust reaching the Alpine chain during spring in 2014. Elements found in dust were classified as a function of their origin and compared with Saharan sources. A strong enrichment in Fe was observed in snow containing Saharan dust. In our case study, impurities deposited in snow anticipated the disappearance of snow up to 38 days for the 2015/2016 season that was characterized by a strong dust deposition event, out of a total 7 months of typical snow persistence. During the other seasons considered here (2013/2014, and 2014/2015), the advancement in snow melt-out day was 18 and 11 days respectively. We conclude that the effect of the Saharan dust is to anticipate the snow melt-out dates, that is known to have a series of feedback effects: earlier snowmelt can propagate into altered hydrological cycle in the Alps, higher sensitivity to late summer drought, impact on vegetation phenology and carbon uptakes from the atmosphere.

## 1  Introduction

Mineral dust (hereafter referred as dust) plays an important role in Earth's climate and in biogeochemical cycles (Mahowald et al., 2013; 2010; Thornton et al., 2009). It provides nutrients such as iron, nitrogen and phosphorous to marine and terrestrial ecosystems (Aciego et al., 2017; Jickells, 2005; Yu et al., 2015), and it influences the shortwave radiation balance of the atmosphere (Ginoux, 2017; Mahowald et al., 2013). Because of its peculiar optical properties,





dust efficiently scatters incoming solar radiation, and exerts a direct climate forcing in the atmosphere (Tegen and Lacis, 1996). As a function of key variables (e.g. imaginary part of refractive index, height of the dust layer, dust particle size, and dust optical depth), the net radiative forcing of dust can be either negative or positive at the top of the atmosphere (Liao and Seinfeld, 1998; Tegen et al., 1996), representing a significant uncertainty in current climate models (Potenza

5 et al., 2016). The main sources of dust are arid and hyper arid regions of the planet. Under specific atmospheric conditions, fine and coarse particles of dust can be suspended in the troposphere, generating characteristic dust storms (Goudie and Middleton, 2001). These phenomena are not typical exclusively on Earth, but also on other planets of the solar system, as Mars for example (Smith, 2002). Finer dust (< 5μm) has a prolonged atmospheric lifetime, in the order of days, allowing for its long-range transport (Mahowald et al., 2013; Tegen & Lacis, 1996). When dust is deposited on snow-

10 and ice-covered regions its radiative impact at the surface results in a positive radiative forcing (Painter et al., 2012; Skiles et al., 2018). Snow optical properties depend largely on its microstructure and on the presence of impurities (also referred as light-absorbing particles, LAPs), such as carbonaceous or mineral particles (Warren and Wiscombe, 1980). Indeed, dust lowers snow albedo in the visible wavelengths, enhancing the absorption of solar radiation (Di Mauro et al., 2015; Painter et al., 2007), and thus triggering the snow-albedo feedback (Hansen and Nazarenko, 2004). The alterations of the

15 optical properties of snow are known to accelerate the melting processes (Drake 1981; Painter et al. 2012).

The impact of dust on snow melting has been largely investigated in the Western US (Painter et al., 2012; Reynolds et al., 2013), where both radiative and hydrological effects have been assessed (Skiles et al., 2012) using aerial, satellite and Automatic Weather Station (AWS) data (Painter, et al., 2012; 2013; 2018). In this area, the proximity of arid regions to the mountain ranges allows massive dust depositions on snow covered mountain ranges. The snowmelt advancement due

20 to dust depositions ranged from 35 days (Painter et al., 2007) to a maximum of 51 days (Skiles et al., 2012), strongly impacting water supplies around the area (Painter et al. 2012; 2018). First estimations of the impact of dust on snow date back to the beginning of the last century: Jones (1913) estimated one month of anticipated snow melting due to dust deposition in the US. Drake (1981) used a model to estimate the advanced melting of snow next to an active mine, and estimated 4 days of advancement in the snow melt. These advances in snow melt-out dates have important implications

25 on water supply operations, also considering that the runoff from the Colorado River supplies water to over 30 million people in seven US and Mexico (Painter et al., 2012). Increases in dust deposition has been recently observed in this area, and they were linked to human activity and climate change (Neff et al., 2008). Other studies, conducted in Iceland (Dagsson-Waldhauserova et al., 2015; Wittmann et al., 2017), in Himalaya (Gautam et al., 2013), in Norway (Matt et al., 2018) and in the European Alps (Dumont et al., 2017; Di Mauro et al., 2015; Tuzet et al., 2017) reported significant

30 impacts of dust on snow optical properties and snowpack dynamics. Impacts on glaciers optical properties and mass balance were also reported in the literature (Gabbi et al., 2015; Di Mauro et al., 2017; Oerlemans et al., 2009).

The composition of dust varies as a function of its origin (Krueger et al., 2004) and timing (Kumar et al., 2018), with effect on its optical properties (Caponi et al., 2017). Lawrence et al. (2010) presented a comprehensive characterization of the mineralogical and geochemical properties of dust deposited from the atmosphere in the San Juan Mountains

35 (Colorado, US). In this area, dust is dominated by silt and clay particles, indicating a regional source area. In the European Alps, a large fraction of dust reaching high mountains and glaciers is originated from the Saharan desert (Haeberli, 1977; Kandler et al., 2007; Krueger et al., 2004; Schwikowski et al., 1995; Thevenon et al., 2009), but input from local sources cannot be excluded. Even though the Alps are located at a distance of about 3000 km from the largest desert of the planet, they are frequently affected by dust depositions. Thanks to their considerable elevation they act as an orographic barrier,

40 enhancing cloud formation, precipitation and dust scavenging from the atmosphere to the ground (De Angelis and



Gaudichet, 1991; Prodi and Fea, 1979). Dust deposition in the Alps is a well-known process, and its frequency is studied using ice cores from mountain glaciers (De Angelis and Gaudichet, 1991; Thevenon et al., 2009). Each year, Saharan desert provides up to 760 millions of tons of dust to the atmosphere (Callot et al., 2000). Dust reaching Europe is dominated by silicates and aluminium oxide (Goudie and Middleton, 2001), other contributions come from quartz,

calcium-rich particles, sulfates, hematite, and soot (Kandler et al., 2007). The optical properties of particles are directly related to dust composition (Linke et al., 2006), and hence the latter is expected to modify dust radiative effect on snow (Reynolds et al., 2013). Several studies characterized the optical properties of dust and iron oxides (e.g. hematite, goethite etc.) contained in it (Caponi et al., 2017; Formenti et al., 2014). The effect of iron oxides on light absorption was found to be comparable to black carbon (Alfaro et al., 2004), which is known to have important light-absorbing properties (Bond

et al., 2013).

Seasonal snow represents a fundamental reservoir of fresh water in mountain ranges and polar regions. Recent climate changes showed to exert a strong impact on the duration of snow cover (Vaughan et al., 2013), in particular in the European Alps (Beniston, 2005; 2018). It has been observed that, especially in spring, snow cover extent has decreased in the Northern Hemisphere (Brown & Robinson, 2011; Brown et al., 2009). Earlier snow melt can have an impact on

vegetation phenology (Steltzer et al., 2009) and water availability (Beniston et al., 2003), and it is expected to alter hydrologic regimes in the future. Accelerating snow melting due to dust can alter also surface hydrology in large mountain chains like the European Alps. In the Po plain for example, the most important renewable energy source is represented by hydropower. Meltwater from seasonal snow is a fundamental resource for agriculture during spring and summer (Huss et al., 2017). Early snow melting in spring directly impacts the water availability during summer.

Saharan dust can serve as a nutrient for many alpine ecosystems (Field et al., 2010; Okin et al., 2004). At the moment, the impact of Saharan dust events on the biogeochemistry of ecosystems in the European Alps has been poorly analysed (Avila and Peñuelas, 1999). Aciego et al. (2017) recently showed that dust transported from Asia to Western US provides nutrients to montane forest ecosystems. This aspect has never been evaluated for mountain ecosystems in the European Alps, where dust may compete with fine debris from local rocks in providing nutrients to soils. Conversely, the direct

deposition of dust on plants can limit the photosynthetic capacity (Neves et al., 2009), so complex feedback may be involved in the role of dust events in alpine areas. Steltzer et al. (2009) reported results from a manipulation experiment conducted in Western US to study the dependence of vegetation phenology on snowmelt. They measured an advancement of 7 days in snow melt when dust was manually added to the snowpack. This process can simulate a dry deposition from the atmosphere. In the Alps, most of dust depositions occur via wet deposition (mainly snowfalls in high altitude

mountains) (Sodemann et al., 2006), so dust is expected to be included within ice grains. Flanner et al. (2012) showed that when black carbon is internally mixed in ice grains, its radiative effect is stronger. If this holds true also for dust, wet deposition of dust may exert a stronger effect with respect to dry depositions. Shifts in vegetation phenology also impact on timing of migration, breeding, and asynchronies between interacting animal species (Cohen et al., 2018; Thackeray et al., 2016). Dust-induced snowmelt can cause an advancing in the beginning of the growing season, and this can result in

an earlier start in the seasonal cycle of both animals and plants. Changes in snow falls and dust depositions are likely to occur more frequently in a warming climate.

In this paper, we quantitatively estimate the impact of dust from Saharan desert on snow dynamics. As a test area we use the experimental site in Torgnon (Aosta valley, Western Italian Alps) equipped with several sensors for measuring snow properties. Snow dynamics were simulated with a multi-layer, physically based energy balance model (Crocus, Vionnet

et al. 2012), which can incorporate the effect of LAPs in snow and estimate its impact on snow melting (Tuzet et al.





2017). Observed and simulated snow variables are compared and the role of impurities in accelerating the snow melting is discussed. Furthermore, we present a geochemical characterization of dust reaching the Alps, and thus we discuss the possible biogeochemical and hydrological role of dust in the Alps.

## 2    Data and Methods

### 2.1    Torgnon experimental site

The study area is located in the northwestern Italian Alps (Aosta Valley, IT) at an altitude of 2160 m a.s.l. (45°50'40''N, 7°34'41''E). The experimental site belongs to the Phenocam (Torgnon-nd, https://phenocam.sr.unh.edu/webcam/), ICOS (IT-Tor https://www.icos-ri.eu/) and LTER (lter_eu_it_077, https://data.lter-europe.net/deims/site/) networks. The area is a subalpine unmanaged pasture classified as intra-alpine with semi-continental climate. The site is generally covered

10    by snow from the end of October to late May. Further information regarding the site can be found in Galvagno et al. (2013). An Automatic Weather Station (AWS) was installed in 2009 at the experimental site of Torgnon. Air temperature and snow height are measured by a HMP45 (Vaisala Inc.) and a sonic snow depth sensor (SR50A, Campbell Scientific, Inc.), respectively. Albedo is measured with a Kipp and Zonen (CNR4 net radiometer). Snow Water Equivalent (SWE) is measured with a Gamma MONitor (GMON, Campbell) sensor. Data are available at hourly time resolution.

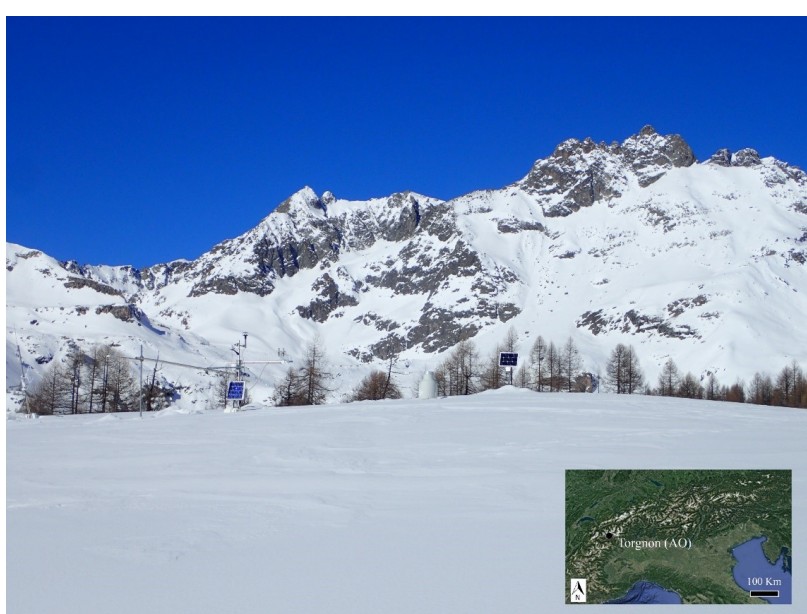

**Figure 1: A picture of the experimental site of Torgnon (AO) in the Aosta valley (Western Alps). The site is located at 2160 m a.s.l.**

In recent years, the application of Red Green Blue (RGB) digital images has gained a lot of interest in Earth and Environmental Sciences. The availability of low-cost digital sensors (typically compact or reflex cameras) has raised the

20    possibility to install automatic station in impervious or remote areas, with the possibility to collect multiple images during summer and winter seasons. Common applications regard the monitoring of vegetation phenology (Julitta et al., 2014; Migliavacca et al., 2011; Richardson et al., 2007), landslides, glaciers (Jung et al., 2010) and snow (Corripio, 2010;





Dumont et al., 2011; Hinkler et al., 2010; Parajka et al., 2012). Regarding the two latter, using digital cameras researchers successfully retrieved snow albedo and snow cover in alpine areas.

For this study, digital RGB images were collected using a Nikon digital camera (model d5000, also referred as 'Phenocam') installed at the experimental site in 2013 in the vicinity of the AWS. Following Richardson et al. (2007),

the camera was pointed North and set at an angle of about 20° below horizontal. Camera focal length is 33 mm and the field of view is 79.8°. The camera was fixed at 2.5 m above the ground and the same view scene was repeatedly captured. Digital images were collected in Joint Photographic Experts Group (JPEG) format and they have a resolution of 12 megapixels, with three-color channels (namely red, green and blue), at 8 bits of radiometric resolution. The images were collected from 10 am to 5 pm (local time), with an hourly temporal resolution. Exposure mode and white balance were

set to automatic.

A Region Of Interest (ROI) was firstly identified in an approximately flat area to analyze snow evolution. Images were acquired during 2013/2016 hydrological years. Red, green and blue chromatic coordinates were extracted from the selected ROI using the Phenopix R package (Filippa et al., 2016). Then, the Snow Darkening Index (SDI, Di Mauro et al. 2015) was calculated from red and green digital number (DN) as follows in Eq. (1):

$$SDI = \frac{DN_{Red} - DN_{Green}}{DN_{Red} + DN_{Green}} \tag{1}$$

SDI was correlated with the concentration of dust in snow (Di Mauro et al., 2015), and was used to represent the spatial distribution of impurities from space (Ganey et al., 2017; Di Mauro et al., 2017), and from hyperspectral imagery of ice cores (Garzonio et al., 2018). SDI calculated from RGB data collected from an Unmanned Aerial Vehicle (UAV) was found to be correlated with SDI calculated from field spectroscopy data (Di Mauro et al. 2015). This motivated the idea

to monitor dust deposition and resurfacing dynamics using repeated digital images from the camera installed in Torgnon. In this work, SDI was calculated for each available image, then a daily average was calculated. Days with SDI >0 were considered as markers of the presence of dust on snow.

### 2.2 Snowpack modeling

Snow dynamics in Torgnon were simulated using SURFEX/ISBA-Crocus model, hereinafter referred as Crocus. Crocus

is a snow model initially developed for avalanche forecasting, and used for hydrological estimation and for numerical prediction (Brun et al., 1989). Crocus is a one-dimensional multilayer model that simulates the evolution of the snow pack based on input meteorological driving conditions (Brun et al., 1989, 1992). Snow dynamics are represented as a function of energy and mass-transfer in the snowpack, between both the snowpack and the atmosphere, and the snowpack and the ground below (Vionnet et al., 2012). In this study, we used a specific Crocus version using the Two-stream

Analytical Radiative TransfEr in Snow (TARTES) radiative transfer model (Libois et al., 2013) to obtain simulations of snow spectral albedo as a function of snow properties and LAPs concentrations and LAPs optical properties. TARTES model is based on the asymptotic approximation of the radiative transfer theory (AART) (Kokhanovsky and Zege, 2004), and it accounts for the effect of snow microstructure and impurities (dust and black carbon in the study case). Snow spectral albedo simulated with TARTES was used to calculate SDI, and it was compared with SDI calculated from the

digital camera. A complete description of this specific Crocus model version can be found in Tuzet et al. (2017). Crocus is embedded in the SURFEX surface scheme and is permanently coupled with the ISBA-DIF soil model.



Variables needed for running Crocus simulations are: air temperature, direct and diffuse shortwave incoming radiation, longwave radiation, wind speed and direction, specific humidity, surface pressure, solid and liquid precipitation. The model was forced using meteorological data from the station in Torgnon for the seasons 2013/2016 at hourly time step. All variables were measured at the station of Torgnon, except for diffuse shortwave incoming radiation, that was measured

in another station located 2 km far from Torgnon. The instrument used for precipitation measurements (pluvio2 OTT) does not feature a windshield. This can be problematic since underestimations of snow fall can occur during intense wind events. For this reason, we corrected the data following the prescriptions proposed in Kochendorfer et al. (2017). Some manual adjustments to solid precipitations were needed in case of strong wind events. In addition to the above-mentioned meteorological data, the Crocus version of Tuzet et al. (2017) needs dust and black carbon deposition fluxes. In this study,

these fluxes were taken from the atmospheric model ALADIN-Climat (Nabat et al., 2015). For evaluating the impact of dust depositions on the snowpack dynamics, key variables (e.g. albedo, snow depth, snow water equivalent) measured from the AWS were compared with Crocus simulations with and without impurities (dust and black carbon) in snow. In addition, soil temperature was initialized using a spin-up simulation of 4 years.

### 2.3 Dust concentration, size distribution and geochemistry

On April 6[th] 2016, a field campaign was organized to collect snow samples at six different locations placed at few meters from the AWS station. For each location, four snow samples were collected from a pit respectively at a depth of 0, 20, 40 and 60 cm from the surface. Samples were collected using sterilized Corning tubes (50 mL) and kept frozen until successive measurements. Dust concentration and size distribution were measured using a Coulter Counter technique. Samples were melted in a clean room (class 1000 clean room at EuroCold laboratory facilities, University of Milano-

Bicocca) and analyzed with a Multisizer™ 4e COULTER COUNTER®. The instrument was set with a 100 μm orifice, allowing for the detection of particles with a diameter (equivalent spherical) between 2 and 60 μm, divided into 400 size channels. To obtain dust mass from particle volume, a crustal density of 2.5 g/cm³ was adopted. Total dust concentration was calculated considering the integral of the concentration between 2 and 60 μm. Details about the technique can be found in Ruth et al. (2008).

In addition, dust samples collected in the Alps at 150 km from Torgnon (Artavaggio, LC, Italy, 1650 m asl) in March 2014 are used here to characterize the bulk composition of dust events and the elemental input to Alpine ecosystems. These samples are considered representative for those deposited in Torgnon because the main source area of Saharan dust events reaching the Alps is represented by North Algeria (Potential Source Area in Northern Africa 1, PSANAF-1 in Formenti et al. (2011)). Saharan dust events are regional episodes that move large quantities of mineral dust from arid

region to different latitudes and longitudes. There are two main pathways for the transport of dust: it can reach Europe over the Mediterranean and also by looping back over the Atlantic (Israelevich et al., 2012; Sodemann et al., 2006). For this reason, we can assume that the bulk geochemical composition of dust events occurred on different location in the Alps and at different times is comparable. Between February 2014 18th and 20th a relevant event was observed, involving not only Southern Europe and the Alps, but also a large fraction of Europe. It was described as one of the most intense

events of this kind observed in the last years. The event was associated to a particularly favorable atmospheric setting which could uplift a massive amount of Saharan dust from North Africa and transport it toward Europe in association to southwesterly winds driven by an anticyclonic structure located on the Central Mediterranean. Given the magnitude of the event, many studies reported it, spanning from microbiology (Meola et al., 2015; Weil et al., 2017), to remote and proximal sensing (Dumont et al., 2017; Di Mauro et al., 2015; Tuzet et al., 2017), and atmospheric chemistry and physics

(Belosi et al., 2017; Telloli et al., 2018). Snow samples were transported before melting in a cold facility, where they




were stored until the preparation for the successive analyses. At first, they were melted and an aliquot (5-10 mL) was measured through Coulter Counter technique (CC) for the determination of dust size distribution. These data were already published (Di Mauro et al., 2015). A second aliquot consisting in few mL of melted snow, was dedicated to Instrumental Neutron Activation Analysis (INAA) for the analysis of elemental composition (Greenberg et al., 2011). To this aim, dust
was extracted and separated using a filtration system, equipped with polycarbonate membranes (pore size 0.4 μm, well below the typical volume mode grain size of Saharan dust deposited on the Alps). Two distinct samples were prepared. One sample (SH1) was extracted from the reddish snow corresponding to the snow deposited during the Saharan event; it consisted in $7.2 \pm 0.2$ mg of dust. A second sample (SH2) was prepared for comparison filtering clean white snow. In this case, given the low concentration of impurities, it was possible to retrieve only $202 \pm 11$ μg of particulate matter. For
both samples, in addition to absolute concentration (mass fraction), also normalized ones were calculated. The average upper continental crust composition (UCC, Rudnick and Gao 2003) was selected as normalizing reference to highlight the influence played by crustal-derived material and the possible role of non-crustal sources for specific elements. Neutron irradiation was performed at the LENA laboratories at the University of Pavia (Borio di Tigliole et al., 2010), where a TRIGA Mark II research nuclear reactor is installed (250 kW). Activated samples were successively analyzed using high
purity Germanium detector available at the Radioactivity Laboratory of the Milano-Bicocca University. Two irradiations and several acquisitions of the γ-spectra were necessary to detect the largest number of radionuclides, ranging from the short-lived species to the long-lived ones. For a complete description of the method, see Baccolo et al. (2015, 2016).

**2.4     Dust transport and deposition modelling**

In addition to the ALADIN-Climate model, dust transport and deposition were monitored using the NMMB/BSC-Dust
model. This is an online multi-scale atmospheric dust model (Pérez et al., 2011), it was used here to provide dust forecasts from the Saharan desert to the European Alps. NMMB/BSC-Dust provides both atmospheric concentration and deposition fluxes of dust with a 0.3º x 0.3º horizontal resolution. During the three seasons considered here, we classified dust events in 'strong' and 'weak'. 'Strong' events were characterized by dust deposition fluxes larger than 800 mg/m², 'weak' events featured lower concentrations. The timings of the events simulated with the NMMB/BSC-Dust model were qualitatively
compared to those simulated with the ALADIN-Climate model during the period analysed here (2013/2016).

**3     Results and Discussion**

**3.1     Dust depositions**

The period between 2013 and 2016 was characterized by two 'strong' events (dust fluxes > 800 mg/m²), and several 'weak' events (dust fluxes < 800 mg/m²) distributed during the seasons. The 'strong' events occurred on February 2014
and on April 2016. The event of February 2014 was already analysed in the scientific literature (Di Mauro et al. 2015; Tuzet et al. 2017; Dumont et al. 2017). The event of April 2016 lasted several days and transported a considerable dust to the Western sector of the European Alps (Fig. 2) (Greilinger et al., 2018). According to NMMB/BSC-dust model, during these two "strong" events, most of dust was deposited in the Alpine chain mainly via wet deposition. In Figure 2, we show an example (5th April 2016) of the concentration of dust deposited according to NMMB/BSC-dust, and a
longitudinal and latitudinal transect. NMMB/BSC-dust predicted up to 1600 mg/m² of dust deposition in Western Alps. The complex topography of this region probably acted as an orogenic barrier promoting the condensation of water vapor containing dust particles. This process generated the characteristic "red snow" often observed in the Alps (De Angelis & Gaudichet, 1991). In the latitudinal and longitudinal profiles, it is clearly visible that the plume reached almost 6 km in altitude. The highest concentrations in the atmosphere were reached in South France and in the North-West of Italy. The



experimental site of Torgnon is located in Italian Western Alps, and it represents a good candidate for analysing the effect of these strong dust events on snow dynamics.

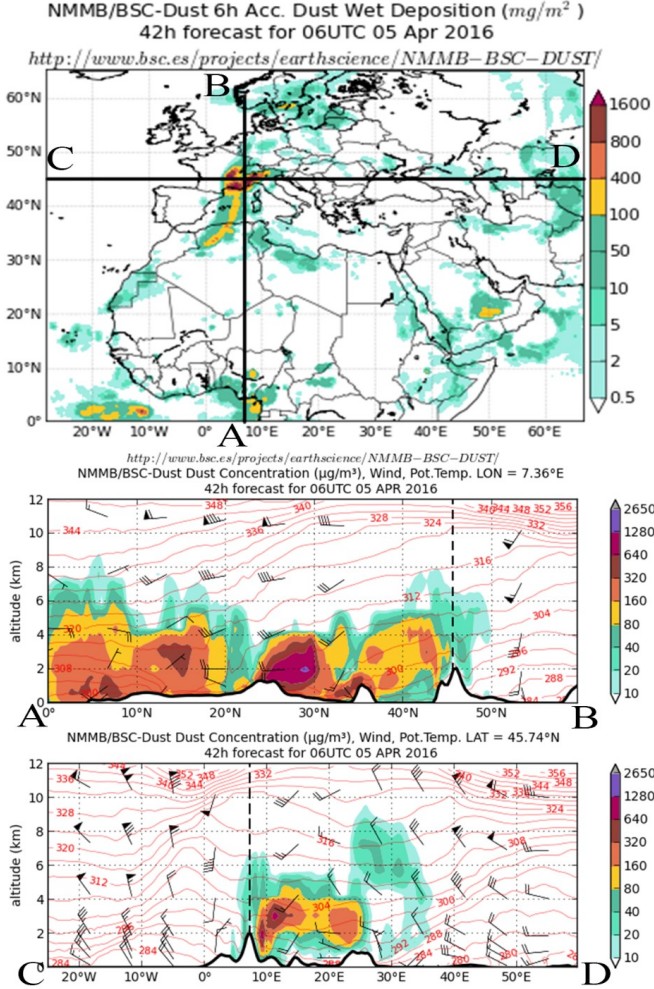

**Figure 2 NMMB/BSC-Dust forecast for the event of April 2016. In the top panel, the estimated surface concentration via wet deposition. The lower panels represent respectively a latitudinal and longitudinal transect centred on the city of Aosta (45.74N; 7.36E). Images from the NMMB/BSC-Dust model, operated by the Barcelona Supercomputing Center (http://www.bsc.es/ess/bsc-dust-daily-forecast/).**

### 3.2    Observed and simulated snow dynamics

In this section, we present data from the station in Torgnon, and simulations made with Crocus model. In Figure 3, a comparison of variables observed and simulated using impurity fluxes is presented. We show time series of snow albedo, snow water equivalent (SWE), and snow depth (SD). In general, Crocus model well represented snow dynamics during 2013/2016 hydrological seasons. In Table 1, we present a quantitative comparison (coefficient of determination, $R^2$, and Root Mean Square Error, RMSE) between snow variables observed and simulated considering the effect of impurities.





Beside a general agreement between observed and simulated variables, it must be noted that some mismatches were observed. For example, snow albedo was underestimated from Crocus during the accumulation period (see Fig. 3a). Instead, during the melting period, the decreasing trend observed in snow albedo was well reproduced by Crocus model accounting for the role of impurities. The albedo simulated by Crocus without impurities was always higher the one
simulated accounting for impurities. The comparisons between observed and simulated SWE and snow depth show a large interannual variability. SWE is strongly overestimated in the 2013/2014 season; while during the accumulation period snow depth is well represented in the model, the melting rate is higher in the observed snow depth. This results in a delay of snow melt-out dates in both Crocus simulations (with and without impurities). A similar pattern in snow depth is observed also in the 2014/2015 season. Unfortunately, measured SWE was not available for this season. During the
2015/2016 season, the correlation between the observed and simulated snow depth accounting for the impact of impurities was very high both for snow depth ($R^2$ = 0.96 RMSE = 0.04 m) and SWE ($R^2$ = 0.97; RMSE = 13 mm). The difference in snow melt-out dates between observed and simulated data accounting for LAPs was 12, 10 and 11 days, respectively for the 2013/2014, 2014/2015, and 2015/2016 seasons. Instead the comparison between snow melt-out dates simulated with and without impurities was 18, 11, and 38 days for the 2013/2014, 2014/2015, and 2015/2016 seasons, respectively.

In Figure 3, we show also a qualitative comparison between the dust fluxes simulated with ALADIN-Climate and with the NMMB/BSC-dust model. "Strong" and "weak" dust deposition events simulated with the NMMB/BSC-dust model are represented as large and small stars, respectively. ALADIN-Climate fluxes are reported as well. In general, a good agreement between the two models was observed. The two most intense events (February 2014, and April 2016) are identified by both models. Smaller events are also reproduced, whereas sometimes small events are seen only by
ALADIN-Climate.

Once dust fluxes are deposited on the snowpack, they are buried by new snowfalls. In Figure 3e, we show the multilayer concentration of dust in snow simulated with Crocus. It is clear that dust is resurfaced at the end of the season, when the snow albedo feedback intensifies, and promotes the melting. The surface concentration of dust (average of the first 10 cm of snow) in the three seasons considered in this study show an important interannual variability (Fig. 3f). In fact, whereas
the first two seasons shows surface concentrations of dust lower than 150 µg/g, the last season (2015/2016) shows concentrations up to 350 µg/g at the end of the season. This also explains the large change (38 days) in the snow melt-out dates observed in the data.





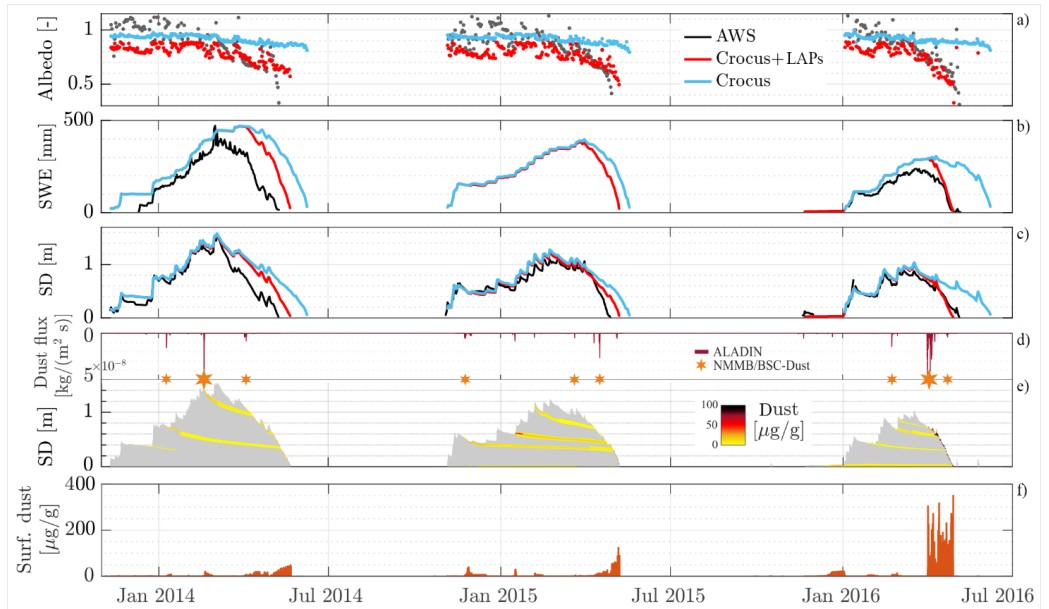

**Figure 3 a)-b)-c): time series of albedo, snow water equivalent (SWE), and snow depth (SD) measured with the AWS and simulated with Crocus model including and excluding the impact of LAPs. d): dust fluxes simulated with ALADIN (bars), and strong (large stars), and weak (small stars) dust events simulated with NMMB/BSC-Dust. e): dust concentration in the**
5 **snowpack simulated with Crocus superimposed on the snow depth profile (shaded area). f): surface concentration (first 10 cm) of dust simulated with Crocus.**

|  | 2013/2014 | 2014/2015 | 2015/2016 | All years |
|---|---|---|---|---|
| **SD [m]** | $R^2 = 0.84$ | $R^2 = 0.9$ | $R^2 = 0.96$ | $R^2 = 0.93$ |
|  | RMSE = 0.15 | RMSE = 0.08 | RMSE = 0.04 | RMSE = 0.08 |
| **SWE [mm]** | $R^2 = 0.63$ | / | $R^2 = 0.97$ | $R^2 = 0.96$ |
|  | RMSE = 73.3 | / | RMSE = 13 | RMSE = 23.3 |
| **Albedo [-]** | $R^2 = 0.74$ | $R^2 = 0.6$ | $R^2 = 0.6$ | $R^2 = 0.72$ |
|  | RMSE = 0.02 | RMSE = 0.02 | RMSE = 0.04 | RMSE = 0.02 |

**Table 1 Comparison between snow depth (SD), snow water equivalent (SWE) and albedo observed from the AWS station in Torgnon and simulated with Crocus accounting for the impact of impurities.**

10  Results from samples collected in Torgnon showed that significant concentrations of dust were present in the snowpack in April 2016 (Figure 4). It is interesting to note that the mode of the dust size distribution is 7.9 µm for surface snow, 8.5 µm, and 8.5 µm for snow samples collected at 20 cm and 40 cm respectively, instead snow sampled at the bottom of the snowpack (60 cm depth) features a mode of 3.2 µm. This deeper layer can be due to eventual scavenging of small dust particles by meltwater. The first three distributions can be due to the 'weak' depositions happened in February and

15  March, and then buried by new snow. At the bottom of the snowpack, finer particles were found. Dust size distributions are compatible with other measurements of dust enclosed snow and ice in the Alps (Maggi et al., 2006), and in Caucasus (Kutuzov et al., 2013). Samples showed in Figure 4 feature a significant noise in the tail of the distribution. This can be ascribed to the aggregation of fine particles or to an input of local larger particles. Total concentration of dust in Torgnon was estimated by adding up different channels from the size distributions. Among the six different snow profiles




measured, surface concentrations reached a maximum of 65 $\mu g_{dust}$ g$^{-1}_{snow}$ with a mean of 45.6 $\mu g_{dust}$ g$^{-1}_{snow}$ and a standard deviation of 15.8 $\mu g_{dust}$ g$^{-1}_{snow}$.

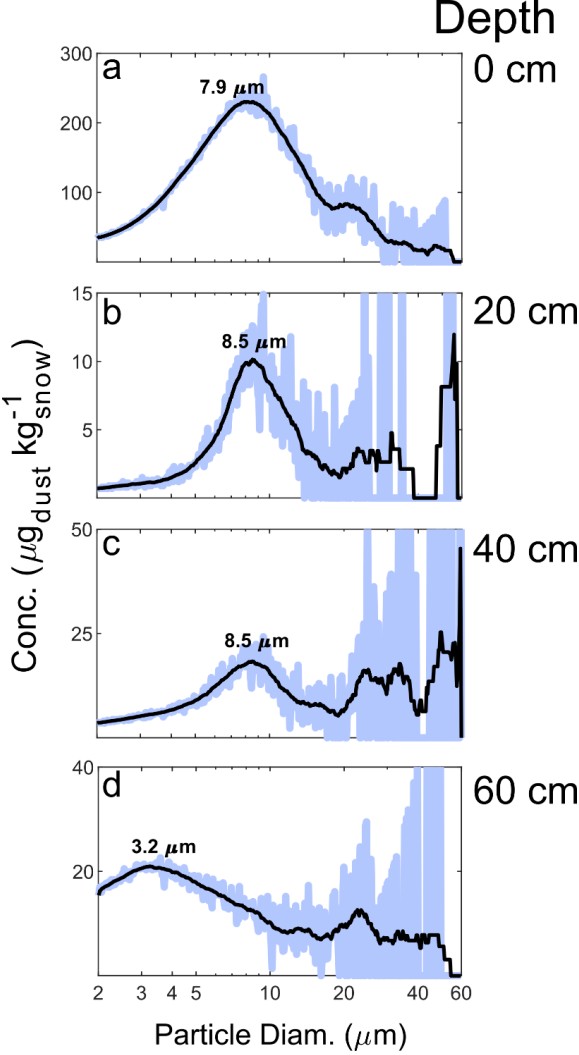

**Figure 4 Dust particles distribution (expressed in $\mu g_{dust}$ kg$^{-1}_{snow}$) for a snow profile sampled at Torgnon on April 6[th] at different depths (0 cm, 20 cm, 40 cm, and 60 cm). Blue lines are experimental data, black lines are moving averages (kernel: 25 points). Numbers in the plots represent the peak of the size distributions. Please note that the scale is changing within different plots.**

Measurements of dust concentrations are available only for April 6[th] 2016. On this day, the dust concentration profile simulated by Crocus spans from 11 (bottom) to 108.7 (top) $\mu g_{dust}$ g$^{-1}_{snow}$. Modelled and measured surface concentrations of dust showed some difference: 43.7 $\mu g_{dust}$ g$^{-1}_{snow}$ for the most concentrated surface sample, and 65.1 $\mu g_{dust}$ g$^{-1}_{snow}$ for the mean of the six snow pits. This variability can be explained by the strong spatial mismatch between the spatial resolution of ALADIN-Climate model (50km) and the punctual measurement of dust concentration. Differences can also depend on snow sampling vertical resolution and Crocus layer thickness. Model improvements are needed to downscale the spatial



resolution of LAPs fluxes. The installation of wet and dry sampler (e.g. deposimeter) at experimental sites may help to drive Crocus model with measured deposition fluxes. It is important to notice that ALADIN-Climate predicted also depositions of black carbon. At the moment, we do not have measurements to validate this estimation, but the presence of black carbon in snow may have amplified the snow-albedo feedback in the snowpack.

Hereafter we focus on the 2015/2016 season, since Crocus simulations with impurities resulted in a 38 days advancement of the snow melt-out date compared to the corresponding simulations without impurities. This season was characterized by dust surface concentration in snow almost double with respect to the other two seasons considered in this study (see Fig. 3f). In Figure 5, we show the comparison of the snow depth simulated with Crocus including and excluding the impact of impurities. During the 2015/2016 season, about one meter of snow was on the ground in Torgnon. The model

without impurities predicted a longer persistence of snow on the ground than the model with impurities. Two late snowfalls occurred in May, and this probably increased the difference between the simulations. Since air temperatures were still close to 0° (data not shown), snow was preserved at the ground in the simulations without impurities, and this further prolonged the snow season duration. The presence of impurities induced an advancement of the disappearance of snow in Torgnon. Considering that first significant snowfalls occurred in January, the snow season was shortened of

about 20% of the total because of impurities. In Figure 5, we also plot SDI index calculated from the radiative transfer model (TARTES) included in Crocus (SDI-Crocus hereafter), and from the RGB camera (SDI-Phenocam hereafter). Regarding the digital camera data, days with SDI >0 are represented as shaded green bands. We observed an agreement between the two data set. SDI-Crocus showed an increasing trend during April. In particular, at the beginning of April two peaks in SDI-Crocus are seen also from SDI- Phenocam. A peak then is not clearly seen by the digital camera, this

could be due to the occurrence of two small snowfalls during the resurfacing of dust layers. At the end of April, the concentration of dust on the surface of snow is well represented both by Crocus and digital images. During this last period, the accumulation of dust on snow further increases light absorption and decreases the albedo. A marked change in snowmelt rate is observed around the 20th of April. This further induced an earlier snow melt-out, which was comparable with that observed from AWS data (see Fig. 3e). The agreement between SDI-Crocus and SDI- Phenocam suggests that

low cost digital RGB data can be used for monitoring the resurfacing of dust in snowfields, useful for satellite and model validation. In order to use quantitatively these RGB data, further comparisons with field spectroscopy and ground data are needed.





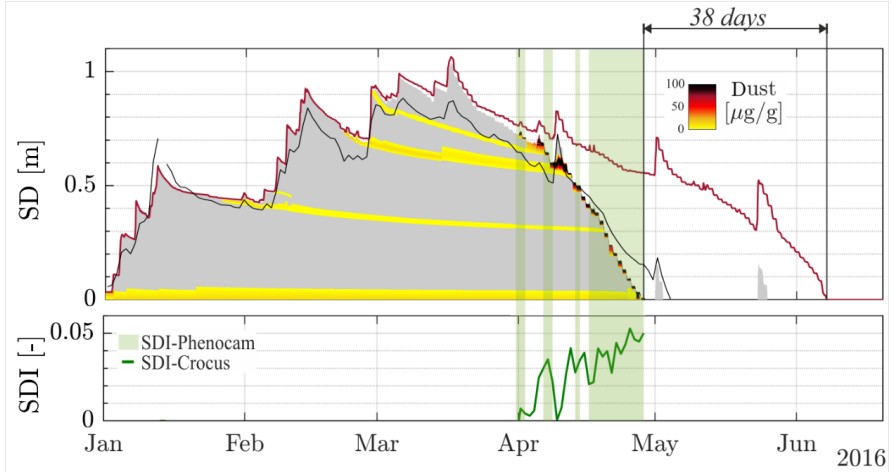

**Figure 5 Comparison between snow depth simulated using Crocus with impurities (grey area) and without impurities (purple line). Observed data are also showed (black line). Dust concentration in snow is represented. Shaded green bands represent days with SDI-Phenocam >0. SDI-Crocus is represented as a continuous green line.**

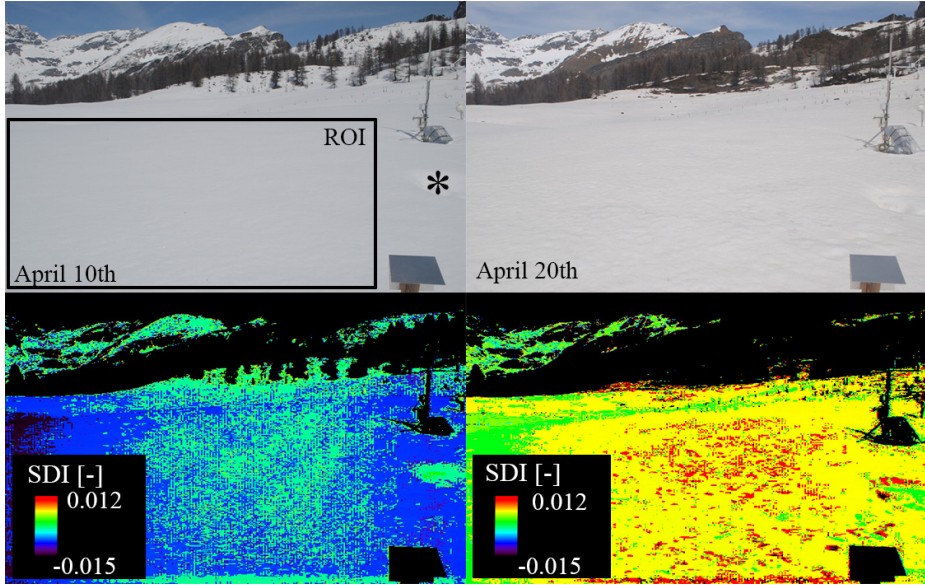

**Figure 6 Top panels: examples of digital images acquired from the Phenocam installed at Torgnon before and after the resurfacing of dust layers. Bottom panels: Snow Darkening Index (SDI) calculated using the Red and Green channels of the images. A region of interest (ROI, in the top left panel) was used to create SDI time series. The asterisk in the top left panel indicates the position of the snow pit.**

In Figure 6, we show two examples of digital images collected from the digital camera. Spatial variability of SDI can be explained by local topography. The experimental site is located in a plain area, with a gentle slope (~5°). Microtopography created by snow melting and refreezing cycles can locally concentrate and dilute impurities in the snow field, also in relation to the differential sun exposure. Surface runoff may also represent an important process in shaping snow surface and in distributing dust in snow fields. This may explain the variability observed in SDI and also the differences between the measured dust concentration in snow samples and Crocus modelled concentrations. Dust redistribution on snowfields



might strongly affect its radiative impact. In Figure 6, we present two SDI maps acquired before (April 10[th]) and after (April 20[th]) the resurfacing of dust layers. The transition from cold to warm colours reflects the increase in the values of the index. Positive values of SDI are here associated with the presence of dust on snow (Di Mauro et al. 2015). On the right of both images, a snow pit is visible. It is interesting to note that in the SDI map from April 20[th], a red layer is visible
in the snow pit. This can be associated with the precedent 'weak' depositions from February and March, which were concentrated in a thin snow layer by melting during early spring. At the end of the season, 'weak' and 'strong' depositions are concentrated by surface melting. This process amplifies the feedback mechanism of dust on snow. In fact, while the melting of snow concentrates the dust on the surface, higher concentrations of dust intensify the melting. This feedback is expected to act for each day with sufficient solar radiation during the output phase. The feedback is also expected to be
enhanced until the total disappearing of the snow cover.

SDI is also sensitive to other impurities such as black carbon (Di Mauro et al., 2017) and organic material (Ganey et al., 2017). We cannot exclude that other impurities were present on snow surface, but at present we do not have enough data to evaluate these aspects. We interpret SDI variability only in relation to dust deposition and resurfacing. In the selected ROI (upper-left panel in Figure 6), frequency distribution of SDI shows a peak at 0.005. Inverting the nonlinear model
developed in Di Mauro et al. (2015), this SDI value is associated with dust concentration equal to 56 $\mu g_{dust}$ $g^{-1}_{snow}$. This value is very close to the concentrations measured with the Coulter Counter integrating particles smaller than 60 $\mu m$, that reached a maximum of 65 $\mu g_{dust}$ $g^{-1}_{snow}$.

Our estimations of shifts in melt-out days are comparable with previous findings in the Western US, where it was estimated that the presence of mineral dust in snow determines a reduction of snow cover up to 51 days (Painter et al.,
2007; Skiles et al., 2012). The interesting point is that, despite the distance from dust sources is larger in the Alps than in the US, the advancement of the snowmelt owing to dust is comparable, at least for a season (2015/2016) with a major Saharan dust deposition. Tuzet et al. (2017) estimated up to 9 days of advanced snowmelt during 2014 in a lower altitude site located in the European Alps as well. In this paper, we estimate an advance in snow melt-out days of 18, 11, and 38 days for the three seasons considered. The estimation for the season 2015/2016 is very high, considering also that snow
cover normally lasts about 7 months at this altitude (2160 m). Our estimation may be affected by the overestimation of impurity deposition fluxes from ALADIN-climate model. This could be due mainly to the spatial mismatch between the atmospheric and snowpack models. Nevertheless, snow depth simulated considering the impact of impurities are well correlated with observations. In the future, impurities concentration estimated with atmospheric model should be evaluated using ground observation. In this sense, data from in situ spectrometers (e.g. Dumont et al., 2017; Picard et al.,
2016) and repeated digital images can be very helpful. In fact, the concentration of different impurities may be retrieved from spectral reflectance using both inversion scheme of radiative transfer models and spectral indices.

Studies like the one presented here are important since dust transports are frequent events in Europe (De Angelis and Gaudichet, 1991; Collaud Coen et al., 2004; Greilinger et al., 2018; Wagenbach and Geis, 1989). For example, dust transport and deposition to Sierra Nevada (Spain) and Caucasus on March 2017[1] and on March 2018[2] were reported by
the media. Dust transport is a natural phenomenon, but it can be intensified by anthropogenic activities (Neff et al., 2008). Further research is needed to assess possible input of local dust to mountain environments. Recently, dust was found to

---

[1] https://earthobservatory.nasa.gov/images/89772/spanish-peaks-turn-tan

[2] https://www.bbc.com/news/world-europe-43533804



be more important than temperature in determining snow melt in Western US (Painter et al., 2018). In the future, the role of dust and air temperature should be determined also in the Alps.

Although no trends were found in the annual number of Saharan dust days since 1997 (Flentje et al., 2015), further research is needed to assess the role of impurities on snow dynamics in the Alps. Measurements of surface concentrations

of dust and black carbon in snow are very scarce in the whole Alpine chain. At the Jungfraujoch station (3454 m.a.s.l), dust concentration in the atmosphere is measured in continuous (Collaud Coen et al., 2004). A comparison with these data will be fundamental to validate Saharan dust fluxes in the Alps and to quantify their effect on snow dynamics.

Snow duration was extremely short during 2015/2016 hydrologic year. Usually, grassland in Torgnon is covered by a thick snow cover from the end of October to late May (average 1928-2010). During 2015/2016 snow arrived in January

and disappeared at the beginning of May. It is known that earlier snowmelt impacts on the carbon uptake period (Galvagno et al., 2013), altering carbon exchange with the atmosphere during spring. Shifts in phenological dates, such as the beginning and end of season, may impact ecosystem functioning related to net and gross ecosystem productivity in alpine grasslands, and might lead to early depletion of soil moisture and early senescence related to summer water stress. Extreme events like heat waves have impacts on phenology of mountain grasslands (Cremonese et al., 2017). With future

climate changes, these extreme events are likely to increase. With the intensification of climate changes, snow is expected to occur later in autumn and to be depleted earlier in spring (Frei et al., 2018; Verfaillie et al., 2018), with significant impact for the hydrological cycle. The effect of Saharan dust in the European Alps is to accelerate the melt via direct and indirect effect on snow albedo thus enhancing snow season shortening.

### 3.3    Geochemical characterization of dust in snow

Dust composition is strictly tied to its optical characteristics and hence to its radiative effect on snow (Caponi et al., 2017; Reynolds et al., 2013). Iron oxides contained in dust are particularly absorptive in the visible wavelengths (Alfaro et al., 2004; Linke et al., 2006), and this further enhances the albedo feedback when dust is deposited on snow. The composition of dust is important also for the correct representation of dust in radiative transfer models and global climate models (Albani et al., 2014).

Saharan dust events provide an input of nutrients to Alpine ecosystems (Goudie and Middleton, 2001), and this has been poorly studied in the scientific literature (Arvin et al., 2017). Hereafter, we provide results from a geochemical characterization of dust sampled in snow in the Alps (Artavaggio, LC, Italy). INAA allowed detecting 36 elements, spanning from the so-called major elements (the ones whose oxides constitute more than 1 % of the average composition of Earth crust) to many minor and trace ones. Data of interest are showed in Figure 7, the full list of elemental

concentrations is reported in Table 2.




| Element | SH1 | | SH2 | |
|---|---|---|---|---|
| | Conc. (µg/g) | Conc. (UCC norm.) | Conc. (µg/g) | Conc. (UCC norm.) |
| Na* | 0.56(0.05) | 0.47(0.04) | 0.20(0.03) | 0.17(0.03) |
| Mg* | 1.2(0.2) | 0.8(0.1) | 0.36(0.09) | 0.24(0.06) |
| Al* | 6.3(1.4) | 0.8(0.2) | 1.35(0.30) | 0.17(0.04) |
| Si* | 20.0(3.5) | 0.6(0.1) | 4.8(2.5) | 0.15(0.08) |
| K* | 1.7(0.2) | 0.75(0.09) | 0.45(0.04) | 0.19(0.02) |
| Ca* | 1.6(0.5) | 0.6(0.2) | 0.6(0.2) | 0.25(0.09) |
| Ti* | 0.60(0.05) | 1.6(0.1) | 0.09(0.02) | 0.25(0.06) |
| Mn | 470(50) | 0.61(0.07) | 180(33) | 0.23(0.04) |
| Fe* | 4.0(0.4) | 1.0(0.1) | 0.018(0.003) | 0.38(0.05) |
| Sc | 13(1) | 0.91(0.08) | 3.1(0.4) | 0.22(0.03) |
| V | 100(10) | 1.0(0.1) | 26(5) | 0.27(0.05) |
| Cr | 123(27) | 1.3(0.03) | 84(22) | 0.9(0.2) |
| Co | 14(1) | 0.82(0.06) | 6.6(0.7) | 0.38(0.05) |
| Ni | 35(8) | 0.7(0.2) | 38(11) | 0.8(0.2) |
| Zn | 132(13) | 2.0(0.2) | 233(30) | 3.5(0.4) |
| As | 6(1) | 1.3(0.2) | 4(1) | 0.9(0.2) |
| Se | <0.002 | - | 0.5(0.1) | 5(1) |
| Rb | 82(8) | 0.98(0.09) | 28(5) | 0.34(0.06) |
| Sr | 118(19) | 0.37(0.06) | 86(23) | 0.27(0.07) |
| Sb | 1.3(0.2) | 3.1(0.4) | 12(2) | 30(5) |
| Cs | 3.7(0.4) | 0.75(0.08) | 1.5(0.2) | 0.30(0.05) |
| Ba | 500(100) | 0.8(0.2) | 300(200) | 0.5(0.4) |
| La | 43(5) | 1.4(0.1) | 9(2) | 0.29(0.05) |
| Ce | 87(4) | 1.39(0.06) | 29(2) | 0.46(0.04) |
| Nd | 38(8) | 1.4(0.3) | 10(6) | 0.4(0.2) |
| Sm | 7.2(0.9) | 1.5(0.2) | 1.5(0.2) | 0.32(0.05) |
| Eu | 1.5(0.2) | 1.5(0.2) | 0.24(0.07) | 0.24(0.07) |
| Tb | 1.05(0.08) | 1.5(0.1) | 0.22(0.05) | 0.31(0.06) |
| Ho | 1.05(0.08) | 1.3(0.1) | 0.24(0.05) | 0.29(0.07) |
| Yb | 3.9(0.6) | 1.9(0.3) | 0.7(0.2) | 0.3(0.1) |
| Hf | 7.6(0.7) | 1.4(0.1) | 1.5(0.2) | 0.28(0.04) |
| Ta | 1.8(0.5) | 2.0(0.6) | 0.4(0.1) | 0.4(0.1) |
| W | 3.1(0.7) | 16(3) | 19(3) | 100(15) |
| Hg | 0.4(0.1) | 8(2) | 2.9(0.9) | 60(20) |
| Th | 12(1) | 1.2(0.1) | 2.6(0.6) | 0.24(0.05) |
| U | 2.5(0.6) | 0.9(0.2) | 2.2(0.9) | 0.8(0.3) |

**Table 2 The elemental composition of SH1 and SH2. Data are expressed in terms of µg g-1 and are referred to the mass of the extracted material, not to the considered snow volume. For the elements marked by the asterisk, concentrations are expressed in terms of % mass fractions. Normalized concentrations were calculated considering the Upper Continental Crust as a reference** (Rudnick and Gao, 2003)**.**



UCC-normalized concentrations of major elements are shown in Figure 7a. It can be easily appreciated that SH1 and SH2 display a very different composition. SH1, corresponding to the dusty snow deposited during the Saharan advection episode of February 2014, presents a typical crustal signature, with UCC normalized values close to 1. On the opposite SH2 shows very low normalized concentrations, suggesting that in this case the crustal fraction is not the dominant one.

Since all the considered major elements are strongly depleted (normalized concentrations span from 0.17 in the case of Na, to 0.38 for Fe), it can be inferred that probably its composition is dominated by the only major element, which is not considered here: carbon. Unfortunately INAA is not suited for its detection, but it is known that the carbonaceous fraction is an important component of snow impurities (Li et al., 2016; Wang et al., 2015). Comparing SH1 to Sahel and Saharan dust source composition a substantial correspondence can be appreciated, as it is possible to see in Figure 7. This is not

unexpected, but direct observations linking the geochemical properties of Saharan dust to the dust deposited in the Alps are quite scarce.

One of the main differences between SH1 and SH2 regards Iron (Fe). With respect to this element, SH1 presents absolute and relative concentrations that are more than two orders of magnitude higher than in SH2. This suggests that Saharan dust could be important for supplying this essential element to high altitude alpine ecosystems where other nutrient

sources could be limited, as already pointed out to in relation to other species and to other environments (Avila et al., 1998; Greilinger et al., 2018; Rizzolo et al., 2017). Another issue related to Fe concentration in atmospheric dust is related to its optical properties, since iron oxide concentration and mineralogy strongly influence them (Alfaro et al., 2004; Caponi et al., 2017; Formenti et al., 2014; Linke et al., 2006). The large abundance of Fe is thus expected to affect the radiative effects of dust on snow (Reynolds et al., 2013).

Looking at Ca and Ti further information can be inferred about the most likely provenance of SH1. North African sources (grey lines in Fig. 7a) can be clearly distinguished in relation to the content of these two elements, indeed two groups are recognized. A first one is characterized by high Ca concentration and low Ti content, the second groups shows an opposite composition, with a larger amount of Ti and lower one of Ca (see Fig. 7a). This is related to the carbonate content of the African samples. Carbonates are rich in Ca (a constituent of these rocks) and poor in Ti, in relation to their limited content

in accessory and heavy minerals. The first group (high Ca and low Ti) corresponds to the samples collected in Western Sahara, where carbonate rocks are common. On the opposite samples from North Africa display an opposed composition. Comparing SH1 to these groups it is clear that its composition is in accordance with the second group, not with the first one. Its provenance is more probably related to the mobilization of dust from the central sector of the Sahara-Sahel dust corridor, i.e. the Hoggar, Chad and Niger basins (Moreno et al., 2006).

The elemental composition of dust might have also important effect on the biogeochemical cycles of the alpine grasslands. Among the elements listed in Table 2 there are elements such as K, and Ca that are known to be relevant for ecosystem functioning (Sardans and Peñuelas, 2015). K for instance is an important micronutrient regulating primarily the mechanisms that mitigate water stress (i.e. stomatal regulation, hydraulic conductivity and osmotic regulation in the plant cells), and some photosynthetic processes (i.e. enzymatic activity and synthesis of proteins) (Qiu et al., 2018; Sardans

and Peñuelas, 2015). Ca, beside affecting soil pH and improving soil structure, has important effects on ecosystem physiology (Schaffner et al., 2012). There are evidences that K input in terrestrial ecosystems depends on atmospheric depositions (beside management activity and fertilization), which play an important role in regulating vegetation functioning and relief nutrients limitation (Sardans and Peñuelas, 2015). However, the impact of the input of K on ecosystem functions depends on the soil characteristics and by the leaching, and the effect of Ca might depend on the soil



pH. The effect of the triplicate atmospheric inputs of K (and nearly doubled input of Ca) associated to the sample SH1 (Table 2) requires more attention and further studies to understand the feedback of Saharan dust deposition not only on the biophysical properties of the ecosystem, but also on the biogeochemistry.

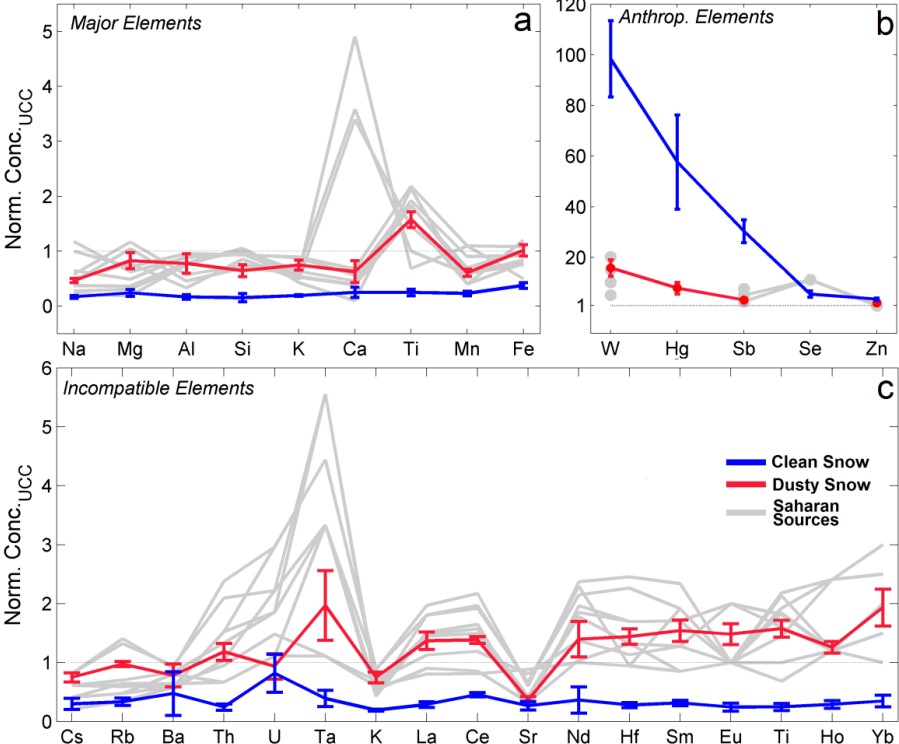

**Figure 7** The elemental composition of the Saharan dust extracted from the snow precipitated on the Alps in February 2014 (red line) and the composition of the particulate matter retrieved from the clean snow deposited few days later (blue line). Grey lines refer to samples collected from the Sahara-Sahel dust corridor (Moreno et al., 2006). They are intended here as references for the potential source areas emitting the mineral dust which is usually transported from North Africa to Southern Europe. Data are expressed in terms of concentration normalized to the average upper continental crust (UCC (Rudnick and Gao, 2003)).
a- Major elements; b- anthropogenic elements, presenting anomalously high normalized concentrations; c- incompatible elements (with respect to Fe), listed following the order proposed by (Sun and McDonough, 1989), therefore Cs and Rb display the maximum incompatibility degree, Ho and Yb the lowest one.

Anthropogenic elements are presented in Figure 7b. This group of elements concerns those elements presenting important positive deviations with respect to UCC composition. They are defined "anthropogenic" to highlight that the their biogeochemical cycles have been strongly impacted by human activities in the last decades and that their mobilization in the environment related to anthropic activities exceeds the natural one (Sen and Peucker-Ehrenbrink, 2012). They are W, Hg, Sb, Se and Zn (Fig. 7b). Again, the signature of SH1 is completely different from the one of SH2. Unlike the case of major and incompatible elements, for anthropogenic elements the sample presenting the higher relative concentration is SH2. SH1 shows values near 1, suggesting that its composition is mainly crustal also for these elements. On the opposite, sample SH2 presents extremely high enrichments, near 100 in the case of W. Such values are not compatible with a crustal origin, contributes from other sources must be involved. Atmospheric emissions related to human activities are the best candidate to explain the enrichment of almost all of these elements. Hg, Sb, Se and Zn are all quite volatile elements, easily mobilized in the atmosphere and related to industrial processes. Given the position of the Artavaggio plains it is





not unexpected to detect such chemical evidences on the Alps. Indeed, the sampling site is located less than 100 km far from the Po valley, one of the most industrialized and densely inhabited regions of Europe. The same interpretation is not sufficient to explain the considerably high amount of W in SH2. There is no previous available information about its occurrence in snow and in general its behaviour in the environment is quite obscure (Koutsospyros et al., 2006). It is

traditionally considered a non-volatile element, given its refractory properties. The high concentration found in SH2 could be related to the anthropic activities characterizing the Po valley, as in the case of the other anthropogenic elements, since W is used in many industrial and manufacturing activities (Koutsospyros et al., 2006). But a different transport mechanism is probably involved. Volatile elements can easily be scavenged from the atmosphere after having be adsorbed on particulate matter (Marx et al., 2008), refractory and non-volatile elements are instead more easily transported directly as

airborne particles generated by industrial processes (Sheppard et al., 2007).

A suite of additional trace element composition is presented in Figure 7c. The elements displayed there were ordered following their incompatibility degree with respect to Fe (Sun and McDonough, 1989). This is a useful geochemical feature to understand the provenance and the geochemical signature of rock related samples. As in the case of major elements an evident distinction concerns SH1 and SH2. SH1 presents a composition which is in perfect accordance with

a crustal origin (normalized concentrations near one), while in SH2 the crustal contribute is definitely secondary, as revealed by the very low normalized concentrations. Focusing on SH1 it can be appreciated that there is a slight enrichment of poorly incompatible elements (the ones on the right side of Fig. 7c). The same feature is also recognized in the African dust sources, as it was extensively discussed by Moreno et al. (2006), which related the point to the geochemical and mineralogical properties of the sources. Sr and Ta are the two elements presenting the most evident

anomalies; a depletion in the first case and an enrichment in the second one. The concentration of Sr is generally related to the presence/absence of carbonates, since Sr is a well-known substituent for Ca in carbonate lattice. In Fig. 7c it is possible to appreciate that SH1 and most of the African sources are significantly depleted in Sr, confirming what already suggested by major elements. Indeed, the samples with low Sr content are the same samples presenting low Ca concentrations, pointing to a limited presence of carbonates and confirming that sources from Western Sahara were not

involved in this episode.

The case of Ta is completely different, given the analytical difficulties related to its detection, its behaviour in the environment is not yet well constrained, but it seems quite common to deal with samples that present an enrichment, in particular when atmosphere-related samples are considered (Filella, 2017). Looking at Figure 7c, it can be seen that both the African sources and to a lesser extent SH1, present a positive anomaly for Ta. Recent studies suggested that the Ta

enrichment in rocks, sediments, and atmospheric particulate matter could be attributed to the effect of chemical weathering. Being Ta extremely stable from a chemical and geochemical perspective, the loss of mobile fractions during weathering, enhanced by atmospheric transport, could explain its enrichment (Baccolo et al., 2016; Vlastelic et al., 2015).

## 4    Conclusions

In this paper, we investigated the role of impurity depositions on snow dynamics. In particular, we analyzed the role of

Saharan dust events on snow melting in a high-altitude site of the European Alps. An overall difference of 38 days was estimated in the disappearance of snow simulated accounting for impurities with respect to snow simulated without accounting for impurities in 2015/2016. During the other seasons considered here (2013/2014, and 2014/2015), the advancement in snow melt-out day was 18 and 11 days. The season of 2015/2016 was characterized by dust depositions almost double with respect to the other years considered in this study. Snow key variables (snow water equivalent, snow





albedo and snow depth) simulated with Crocus model were compared with observed variables from an AWS in the Aosta valley (Western Alps). Good agreement between observations and simulations accounting for the role of impurities was observed. The size distribution of dust found in snow confirms the Saharan origin of the event during April 2016. The geochemical characterization of dust and particulate matter samples distinguished the snow associated to Saharan dust from

clean snow. Dusty snow showed a composition compatible with the geochemistry of the dust sources located in the central sector of the Sahara-Sahel dust corridor, i.e. the Hoggar, Chad and Niger basins North African sources. On the contrary, clean snow was characterized by strong contaminations elated to anthropogenic elements. These results demonstrate that through an accurate geochemical characterization of dust deposited on the Alps, it is possible to identify the different Saharan sources involved in the single transport events. The elemental dataset we presented in this paper could serve as a basis for

assessing the biogeochemical role of dust in snow and in high altitude alpine environments (e.g. enrichment in micronutrients such as K and Ca), and for exploring the interaction between dust composition and its radiative effect on snow.

In the paper, we also made use of repeated digital images for monitoring dust deposition and resurfacing in the snowpack of Torgnon. Dust deposition and resurfacing agreed well with modeling predictions. This allowed us to propose the use of an

RGB index (i.e. snow darkening index, SDI) for tracking dust on snow using repeated digital images from digital cameras. The good agreement between dust deposition and SDI suggests that data from this experimental site can be used as a possible calibration/validation for satellite imagery (e.g. MODIS, Landsat, Sentinel) and for regional and global climate models (WFR-Chem, CLM) validation.

Several questions are still open regarding the role of dust in the Alps. For example, the spatial distribution of dust

concentration on snow at alpine scale has never been quantitatively estimated. Possible differences between Eastern and Western Alps may arise as a function of distance from the sources. Another unresolved issue is the input from local sources: coarser dust particles can be suspended from snow-free areas and deposited on snow. Regarding the geochemical and mineralogical characteristics of dust, future research should explore in detail the relation between dust characteristics and its radiative effect on snow. In addition to the well-known snow-albedo feedback, other complex mechanisms can be involved

in the impact of dust on snow. For example, the presence of dissolved carbonates may accelerate the melt of snow lowering the melting point of snow and ice crystals. The role of carbonaceous particles on snow optical properties in the Alps is also an open question. Measurements of black carbon, brown carbon, organic carbon, and elemental carbon concentration in snow are virtually absent in surface snow in the Alps. The Po plain is one of the most polluted areas of the planet. At lower altitudes, black carbon emissions from fossil fuel combustion and biomass burning may reach snow-covered areas and exert

an impact on snow optical properties. Future research efforts should aim at providing spatially distributed measurements of carbonaceous particles, and this will be a fundamental contribution in the determination of the role of natural and anthropogenic activity on snow melting at regional scale.

**Data availability.** Data used in this paper will be made available upon request to the first author (biagio.dimauro@unimib.it)

**Author contributions.** BDM conceived the idea of the research, analysed the data, and wrote the manuscript with

contributions from all other authors. RG analysed data from the AWS and Crocus. MG, GF, PP, UMdC, and EC established and maintained the experimental site in Torgnon, provided the data from AWS and analysed RGB data from the Phenocam. MD, FT, and ML created Crocus simulations and helped in their interpretation. GB, MC, BD, and VM measured dust concentration and geochemical composition, and helped in their interpretation. MM helped in the interpretation of the geochemical data. MR, SM, EC, and RC supervised the research.



**Acknowledgments.** We acknowledge ARPA (Environmental Protection Agency) of Valle D'Aosta region for maintaining the AWS station in Torgnon and for providing the data set. RGB images were analysed using the Phenopix R package (https://r-forge.r-project.org/projects/phenopix/). The complete set of Phenocam images is available at the following website: https://phenocam.sr.unh.edu/webcam/sites/torgnon-nd/. EC acknowledges the support of the NextData Data-LTER-Mountain project. CNRM.CEN and IGE are part of Labex OSUG@2020. The modelling work was funded by the ANRJCJC EBONI grant n°16-CE01-0006.

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
