# Peer review of "Saharan dust events in the European Alps: role on snowmelt and geochemical characterization"

_The Cryosphere, 2018_

## Short Comment (SC1) · 15 Nov 2018

Hi,

A recent study (Francis et al., 2018) has shown how the changes in the polar jet circulation are favoring the occurrence of dust storms over North Africa and driving the dust transport poleward. Maybe these findings can help to put in context the synoptic situation associated with the events discussed in the current paper.

Francis, D., Eayrs, C., Chaboureau, J.-P.,Mote, T., & Holland, D. M. (2018). Polar jet associated circulation triggered a Saharan cyclone and derived the poleward transport

of the African dust generated by the cyclone.Journal of Geophysical Research: Atmospheres,123. https://doi.org/10.1029/2018JD029095.

Please also note the supplement to this comment:
https://www.the-cryosphere-discuss.net/tc-2018-241/tc-2018-241-SC1-supplement.pdf

**Supplement:**

[supplement omitted: unrelated document]

---

## Referee Comment (RC1) · Kutuzov (Referee) · 15 Dec 2018

The study "Saharan dust events in the European Alps: role on snowmelt and geo-chemical characterization" by Biagio Di Mauro and co-authors is dedicated to a very important topic of impacts of mineral dust on melting of snow in mountainous region. This research is based on observations over 3 years in high-altitude site in European Alps, AWS data and modelling. Additionally authors present a novel and relatively simple technique to monitor dust occurrence on snow. Results of the geochemical analysis of snow sample from one of the dust deposition events were also presented and compared to the chemistry of "clean" snow. Paper is well written and contains a

comprehensive description of the research together with substantial literature review, results and discussion. I recommend this manuscript for publication after a minor revision.

As a general comment in my opinion text could be structured better. In many instances it goes beyond the topic and sometimes the discussion of the methods and results can be found all over the manuscript. Although it is important to mention relevant issues with the methods and data but it is expected that discussion of the results comes after the description of data and method. This complicates reading of the manuscript.

Authors did a great job reviewing a substantial number of previously published researches but the resulted introduction seems excessive and includes a number of repetitions. Some of the statements are repeated later as well. In some instances sentences located in different places are actually stating essentially similar findings and can be combined. I recommend shortening of the introduction and text generally by removing repetitions and information which is not directly related to the conducted research or discussion.

Some specific comments are listed below.

P.1 L.31-34 This was said in previous sentence. Listing of these feedbacks in abstract gives a wrong impression that all these feedbacks were assessed and evaluated here which is not the case. I suggest either to rephrase and generalise or simply to drop it. P2. L21-26 This should be either shortened and moved to line 15 after "The alterations of the optical properties of snow are known …. (…. Painter et al,. 2012)." or removed. P.3 L9. This sentence is then repeated a number of times in the text. Please decide where you want to mention it and remove duplications. Bearing in mind that mineralogy of particles is actually out of the scope of this study. P.3. L.11 This is a bit strong statement (fundamental reservoir). I'd recommend it to be rephrased. Temporal and fundamental do not sound particularly good together. Of course it plays a key role in redistribution and timing of runoff and many other aspects. And then the word fun-

[Figure]

damental is repeated several times later. P.3 L11-19 This paragraph should be moved after the effects of dust ecosystems. And the last sentence (L.35-36) can be placed here "Changes in snow falls and dust depositions are likely to occur more frequently in a warming climate." P3 L.20-36 Please check this paragraph. Order of sentences should be changed so that you first mention what has been done and then point to the knowledge gaps in the Alps. P.4 Fig. 1 The map should be enlarged and zoomed, font size adjusted. Preferably the geochemistry sample site should be included as well. P4 L18-P5 L2 This paragraph should be moved to the introduction. P.6 L5 Which instrument was used to measure diffuse shortwave radiation? P6 L5 These instruments should be listed in site description. P.6 L16 Can you clarify how exactly samples were collected. Was it one sample at depth 20 or one sample for the 0-20 cm layer? What was the total depth of the snow pit? Is there a description/photograph to compare the snow pit with the results of the modelling and dust layers modelling? P.6 L.25 and further. This information is a bit confusing as the plural used for samples of dust which were used to characterize the dust events and elemental input. But at the end of the section we see that there was only one dust (presumably originated in Sahara) sample analysed. This is important issue as it's quite difficult to justify how representative results of analysis of one sample are for other dust events. It should be clearly stated how many samples were analysed for this particular study. Representativeness of this site and samples together with possible dust pathways etc. should be discussed within the results and discussion section rather than here. P.7 L9. "only" is subjective, for some sites this would be considered quite substantial sample. P.7 L18. So far there was no mention about this modelling. Probably it should be mentioned somehow in introduction. P.7 L27 3.1 Modelled dust depositions? P.7 L36-38 repetition. P.8 L2 this strong dust event? (singular?) P.9 L4 than P9. L6 Can you please explain why Crocus model shows a 100 mm SWE in Dec 2013 while this was not observed. Solid precipitation is one of the input parameters isn't it? P10. L.14 This is just one possible explanation though quite doubtful as particles are still quite large to be washed out that simple. It would've been great to see the description (photograph) of the snow pit in 2016 and

to see how it corresponds with modelled structure. Additional samples collected from these dust layers separately could've helped. Another interesting question is how the local mineral particles (rocks, soil, vegetation...) affect snow melting. The mass can be substantial in the snow pack, but of course it will not be modelled by dust deposition model. P.10 L18 The tail in distribution most likely is due to input from local particles. Looking at the photograph there are many rocks and vegetation around the site and Coulter Counter analysis do not distinguish between particles of different nature. This is quite an important issue. If the total mass concentration of mineral particles considered, than highest input would be from the small number of large particles. P.12 L4 Can you please clarify a bit more how exactly BC data were used. Was it an input to Crocus model? How large was the impact compare to dust. Isn't it the largest source of uncertainty? Can the BC signal be separated from the natural dust? Later in the text you mostly discuss the influence of the impurities without specifying. P.12 L17 new paragraph? or maybe it's better to introduce a separate section on SDI P.14 L23 I doubt that it's a good argument to compare advancement in snow melt to distances from the deserts. You can either compare average (long-term) deposition rates or differences in snow duration reduction with similar dust concentrations. P.14 L26 Is it possible to compare bulk concentrations (e.g. CC results) with the deposition modelling results? P.14 L32-35 This is again partly repetition from the introduction. As well as in the next paragraph. Trends are not discussed in this paper at all so it can go to introduction. P.15 L.8-10 This is a bit exaggerated. If the average over 82 years is late May than I believe snow disappeared in early May a number of times. Or maybe not? How many times exactly? So how rare such snow duration actually is? is it really extremely short? The next part of the paragraph is again an introduction-like and can be possibly moved up there too. The importance of the snow duration shifts is explained there. In results section it's better to discuss the exact results. P 16 L.20-27 repetition of the introduction P.20 L. 9-11 I'd suggest to rephrase or remove this. This large topic needs much more regular analysis suitable methodology etc. It just sounds a little bit speculative.

---

## Referee Comment (RC2) · Greilinger (Referee) · 4 Feb 2019

Review of:

**Title: Saharan dust events in the European Alps: role on snowmelt and geochemical characterization**

My recommendation
**Major revisions** due to general and specific comments listed below.

The authors investigate the input of mineral dust (MD) on the geochemistry as well as the impact on snowmelt in the Aosta Valley, Italy at 2160m a.s.l. within the accumulation periods 2013/14, 2014/15 and 2015/16. The study investigate the evolution of snow melt off via in-situ observations, digital images, AWS data and modelling. Besides the investigation of a snow darkening index representative for MD on the snow surface, a geochemical characterization from MD affected and non-affected snow was presented as well. Authors observed a shortening of the snow season, concluding that MD accelerate snow melt-out dates.

The addressed topic is of interest for the Cryosphere community, but also for the climate modelling (e.g. surface albedo feedback) and remote sensing community (e.g. validation and calibration of satellite images).

General comments:
My major concern is the structure and the "red line" throughout the manuscript. The manuscript suffers from many repetitions and the text does not account to the corresponding headline. Therefor it is very hard to read and needs a lot of scrolling to other passages to follow the "story" behind. It would be of much help for the reader and hence also of much more interest if this would be revised and shortened rigorously (therefore major revisions). Authors should think of splitting the results from the discussion into separate section. The manuscript might get ab bit more reader-friendly. Besides, citations seem to be sometimes randomly used whereas they are missing at points were there should be a quote. Authors should cite from recent to past or vice versa, but consistently throughout the manuscript. Details on the general comments raised above can be found in the specific comments below.

Specific comments:

P1 L18 change "snowpack in a..." to "snow packs at a..."

P1 L28 ff Aren't these the results from the comparison of Crocus model results without impurities vs. observations? Otherwise to which reference do the values of 38 days etc. refer?

P1 L34 Include also the importance on snow albedo feedback

P2 L7 remove sentence "These phenomena…"

P2 L13 "…dust lowers THE snow albedo…"

P2 L16 ff Which citation refers to which statement? One reference used twice in one sentence – maybe rewrite the sentence

P2 L22 remove "s" from "century*s*"

P2 L30 maybe also include Greilinger et al here

P2 L34ff remove "of the planet", change "Thanks to.." to "Due to.."

P2 L40 "…precipitation and HENCE dust scavenging…"

P3 L40 define LAPs here

P4 L11ff "…was installed in 2009 measures air temperature (HMP45, Vaisala Inc.) and snow height (ultrasonic sensor SR50A, Campbell Scientific Inc.)."

P4 L18 – P5 L2 belongs to introduction

P5 L3 Include new subsection 2.2 RGB images or digital images or similar

P5 L6 rephrase "…and the same view scene was repeatedly captured" What would you like to say?

P5 L7 …"format WITH a resolution of…AND three-color channels (red, green, blue)…)

P5 L9 Just as suggestion, it is always nice to refer to UTC. If you use local time, please specify time zone.

P5 L16 "Following Di Mauro et al. (2015) and Ganey et al. (2017) SDI was correlated…distribution of deposited impurities …"

P5 L17 What do you mean with "and from hypospectral imagery of ice cores"?

P5 L24 "using THE SURFEX…"

P5 L25 "…estimation AS WELL AS numerical…"

P5 L28 "…and mass transfer between the snowpack and the atmosphere as well as the snowpack and the ground…"

P5 L31 "…snow properties, LAPs concentrations and…"

P5 L33 "…and accounts for…and impurities such as dust and black carbon."

P5 L34 "…TARTES was used to calculated SDI…" How was this done?

P6 L15 "…a few meters apart from the AWS."

P6 L16 "…from a pit at depths of…"

P6 L18 remove "successive"

P6 L20 "…particles between 2 and 60µm (equivalent spherical diameter)."

P6 L22 reference why you use 2.5G/cm³, Why exactly this number?

P6 L25-P7 L17 could be shortened, many passages not necessary. It is the Data and methods section!

P7 L23 "…in 'strong' events with dust deposition fluxes…and 'weak' events with lower concentrations."

P7 L36-P8 L2 removes paragraph, it is the Data and methods section!

P7 L10 remove first sentence

P7 L11 "…variables observed at Torgnon station and simulated with the Crocus model using…"

P8 L15 "In Figure 3d …"

P8 L16ff Remove sentences "Strong and weak…" as well as "ALADIN CLiamte…"

P8 L17 You found a good agreement between the qualitative information, but how about the quantitative?

P8 L27 Please be more explicit why results before explain the large different in snow melt out dates.

P9 Figre3 I personally have difficulties to read and interpret Figure 3d and especially Figure 3e. Maybe explain in more detail in the text (and/or legend) what is the shaded area and what is the colored (reddish, yellowish) area?

P9 Table1 It would be also nice to show the correlation with the Crocus model without impurities

P9 L12 "…8.5µmfor snow samples collected at 20cmand 40cm depth, instead…"

P9 L15 Remove the sentence "At the bottom…"

P9 L16 Authors say that results are comparable with other studies. Please give some numbers what others found, not only the citation.

P9 L17 "Samples shown in…"

P10L8 – P11L27 Please work through the whole section. Parts of the text are already mentioned before, conclusions drawn here are not obvious for the reader. Where exactly do I see the marked change in snowmelt rate and the induced earlier snowmelt in Figure 3e? Here you also mention already some conclusions. It is the Results and discussion section!

P13 Figure5 "…data are also shown (black line)."

P13 L10 "In the upper part of Figure 6 …"

P14 L1 "In the lower part of …"

P14 L5 Why are you sure that the red line is IN the pit? Couldn't this be also a shadowing effect of e.g. an uneven surface? Why should the February event be visible only in the area of the pit?

P14 L14 Which non-linear model? Explain and describe the model of Di Mauro et al. 2015 shortly.

P14 L24-L31 repetition and extensive discussion (maybe start a separate discussion section related to the sections in the results.

P14 L32- P15 L2 Belongs to the introduction

P15 L3-L7 is an outlook, move to summary

P15 L8-L18 another discussion block

P15 L20-L26 move to introduction

P15 L27 "The analysis of the elemental composition allowed…"

P15 L28 Is the threshold of definition of major and minor components referring to > or < than 1% of the average crust composition set by the authors? Reference?

P16 Table2 state somewhere in the legend or in the plot that SH1 is the dust affected and SH2 the clean snow! This would help the reader. Otherwise readers have to go back to the Methods section to check this. What are the value in the brackets? Why are some elements given in % mass fraction and others in µg/g? This makes it difficult to compare.

P17 L1 "Concentrations of major elements normalized to the upper continental crust composition are shown…"

P17 L9 "…see in Figure 7c."

P17 L13 For Fe this is even more than 30 times if I am not mistaken.

P17 L23 remove the sentence "This is related to…" This is discussed again few lines below

P17 L27 remove "not with the first one"

P17 L35 Actually it is not the Ca which is affecting the pH but the related Carbonate! Include here the Carbonate discussion from L23

P17 L37 remove the bracket

P17 L35-P18L3 maybe rephrase the whole paragraph, difficult to see what the authors like to say

P18 Figure7 What is the y-axis in Figure 7b? Remove the sentence "They are intended here…" from the legend. Remove "…, presenting anomalously high normalized concentrations;" Remove everything after "…listed following…" What is meant with "incompatible elements (with respect to Fe)? Indicate here also the nomenclature of SH1 and SH2 to be consistent throughout the manuscript.

P18 L16 include sentence "They are…" already in the first sentence of the paragraph in line 13.

P18 L23 Remove the sentence "Given the position…"

P19 L1-L10 repetition to earlier passages

P19 L12 What does this "incompatibility degree with respect to Fe" reveal? Why use this measure?

P19 L13 Remove the sentences "As in the case..." until "low normalized concentrations." Repetition!

P19 L33 The content of the next section is not a conclusion but a summary! Please stet the conclusion you draw based on your work more explicitly.

P19 L38 "…11days, respectively."

P20 L3 See also http://www.aaqr.org/article/detail/AAQR-18-03-ACPM-0116 to confirm the Sahara dust event. Include citation.

P20 L9 But the fingerprint of the local sources plays also a role. Please state this here in the text.

P20 L20 Maybe you find something in here https://onlinelibrary.wiley.com/doi/abs/10.1034/j.1600-0889.49.issue1.4.x

---

## Author Comment (AC1) · 5 Mar 2019

Dear Dr. Francis,

thank you for suggesting this paper regarding dust poleward transport, we added this reference in the Introduction of the revised version of our manuscript at pg2 ln9.

Line numbers refer to the track-change version of the manuscript.

Best regards,

Biagio Di Mauro and co-authors

---

## Author Comment (AC2) · 5 Mar 2019

Reviewer: Dr. Stanislav Kutuzov

Authors responses are in *italic*, Reviewer's comments are in **bold**. Line numbers refer to the track-changes version of the manuscript.

**The study "Saharan dust events in the European Alps: role on snowmelt and geochemical characterization" by Biagio Di Mauro and co-authors is dedicated to a very important topic of impacts of mineral dust on melting of snow in mountainous region. This research is based on observations over 3 years in high-altitude site in European Alps, AWS data and modelling. Additionally authors present a novel and relatively simple technique to monitor dust occurrence on snow. Results of the geochemical analysis of snow sample from one of the dust deposition events were also presented and compared to the chemistry of "clean" snow. Paper is well written and contains a comprehensive description of the research together with substantial literature review, results and discussion. I recommend this manuscript for publication after a minor revision.**

*Dear Dr. Kutuzov, thank you for the positive evaluation of the manuscript. We have carefully considered each of the Reviewer's comments and suggestions. The Reviewer will find below the responses to general and specific comments.*

**As a general comment in my opinion text could be structured better. In many instances it goes beyond the topic and sometimes the discussion of the methods and results can be found all over the manuscript. Although it is important to mention relevant issues with the methods and data but it is expected that discussion of the results comes after the description of data and method. This complicates reading of the manuscript.**

**Authors did a great job reviewing a substantial number of previously published researches but the resulted introduction seems excessive and includes a number of repetitions. Some of the statements are repeated later as well. In some instances sentences located in different places are actually stating essentially similar findings and can be combined. I recommend shortening of the introduction and text generally by removing repetitions and information which is not directly related to the conducted research or discussion.**

*Thank you for this comment. Following your indications, we shortened the introduction and removed repetitions and information not directly linked to our research.*

**Some specific comments are listed below.**

**P.1 L.31-34 This was said in previous sentence. Listing of these feedbacks in abstract gives a wrong impression that all these feedbacks were assessed and evaluated here which is not the case. I suggest either to rephrase and generalise or simply to drop it.**

*We removed from the abstract the sentences regarding the specific feedback effects of the anticipated snowmelt induced by dust depositions. The sentence now reads:*

*"We conclude that the effect of the Saharan dust is to anticipate the snow melt-out dates, that is known to have a series of hydrological and phenological feedback effects"*

**P2. L21-26 This should be either shortened and moved to line 15 after "The alterations of the optical properties of snow are known . . .. (. . .. Painter et al,. 2012)." Or removed.**

*We moved this paragraph after line 15, and we shortened it. Now it reads:*

*"First estimations of the impact of dust on snow date back to the beginning of the last century: Jones (1913) estimated one month of anticipated snow melting due to dust deposition in the US. Drake (1981) estimated 4 days of advancement in the snow melt. These advances in snow melt-out dates have important implications on water supply operations (Painter et al., 2012)."*

**P.3 L9. This sentence is then repeated a number of times in the text. Please decide where you want to mention it and remove duplications. Bearing in mind that mineralogy of particles is actually out of the scope of this study.**

*We removed the sentence from line 7 to line 10. This topic is then addressed in the discussion in order to link the geochemical composition of dust with its radiative effect when deposited on snow.*

**P.3. L.11 This is a bit strong statement (fundamental reservoir). I'd recommend it to be rephrased. Temporal and fundamental do not sound particularly good together. Of course it plays a key role in redistribution and timing of runoff and many other aspects. And then the word fundamental is repeated several times later.**

*We replaced "fundamental" with "important".*

**P.3 L11-19 This paragraph should be moved after the effects of dust ecosystems. And the last sentence (L.35-36) can be placed here "Changes in snow falls and dust depositions are likely to occur more frequently in a warming climate."**

*We modified accordingly.*

**P3 L.20-36 Please check this paragraph. Order of sentences should be changed so that you first mention what has been done and then point to the knowledge gaps in the Alps.**

*We modified accordingly.*

**P.4 Fig. 1 The map should be enlarged and zoomed, font size adjusted. Preferably the geochemistry sample site should be included as well.**

*We modified accordingly. We also added an aerial view of the site, and the field of view of the Phenocam. Here the new Figure 1:*

[Figure]

*Figure 1: a) location of the experimental site of Torgnon (Aosta), and Artavaggio plains (Lecco) in the European Alps. b) a picture of the experimental site of Torgnon (2160 m a.s.l.). c) aerial view of the site in Torgnon with the location of different instruments installed. The field of view of the Phenocam is also represented with a blue shaded area.*

**P4 L18-P5 L2 This paragraph should be moved to the introduction.**

We shortened the paragraph and we used it to introduce the use of digital images in the methodology section. The sentence now reads:

*"2.2 Digital images analysis*

*In recent years digital images analysis was applied to monitor vegetation phenology (Julitta et al., 2014; Migliavacca et al., 2011; Richardson et al., 2007), landslides, glaciers (Jung et al., 2010) and snow (Corripio, 2010; Dumont et al., 2011; Hinkler et al., 2010; Parajka et al., 2012). Regarding the two latter, using digital cameras researchers successfully retrieved snow albedo and snow cover in alpine areas."*

**P.6 L5 Which instrument was used to measure diffuse shortwave radiation?**

*Diffuse radiation was measured with a BF3 sensor (Delta-T Devices Ltd, Cambridge, UK). We added this information in the text.*

**P6 L5 These instruments should be listed in site description.**

*We added this information in the site description section. Now it reads:*

*"Solid and liquid precipitations were measured with a pluvio2 OTT instrument."*

*We also added:*

*"Wind speed and direction were measured with a CSAT3 three-dimensional sonic anemometer (Campbell Scientific, Inc.)"*

**P.6 L16 Can you clarify how exactly samples were collected. Was it one sample at depth 20 or one sample for the 0-20 cm layer? What was the total depth of the snow pit? Is there a description/photograph to compare the snow pit with the results of the modelling and dust layers modelling?**

*Samples used in this paper were collected from six different snow pits placed at few meters from the AWS station. For each snow pit, we collected a surface samples at 0 cm, and three samples at depths equal to 20, 40, and 60 cm from the surface. The concentrations of dust among different snow pits were very similar, so we presented just an example in Fig. 5. Unfortunately, we don't have high quality photographs of the snow pits for this comparison.*

*Now the sentence reads:*

*"On April 6th 2016, a field campaign was organized to collect snow samples at the experimental site of Torgnon. Six snow pits were dug in different locations placed at few meters from the AWS station. For each snow pit, we collected a surface samples at 0 cm, and three samples at depths equal to 20, 40, and 60 cm from the surface".*

**P.6 L.25 and further. This information is a bit confusing as the plural used for samples of dust which were used to characterize the dust events and elemental input. But at the end of the section we see that there was only one dust (presumably originated in Sahara) sample analysed. This is important issue as it's quite difficult to justify how representative results of analysis of one sample are for other dust events. It should be clearly stated how many samples were analysed for this particular study. Representativeness of this site and samples together with possible dust pathways etc. should be discussed within the results and discussion section rather than here.**

*For the neutron activation analysis, we used two samples: one representative for clean snow and one for the dust event of February 2014 (see line 37 in page 7). We acknowledge that only one snow sample containing dust is not enough to provide a complete overview on the composition of Saharan dust in snow in the Alps, but our analysis may pave the way for a more exhaustive characterization of dust composition in the future.*

*We moved this paragraph to Section 3.3, we also added this sentence (pg 17 ln 31):*

*"For this reason, the dataset presented in this study can be considered representative for the main composition of long-range dust deposition on snow in the Alps."*

*And then in pg 18 ln 11:*

*"We acknowledge that only one snow sample containing dust is not enough to provide a complete overview on the composition of Saharan dust in snow in the Alps, but our analysis may pave the way for a more exhaustive characterization of dust composition in the future."*

**P.7 L9. "only" is subjective, for some sites this would be considered quite substantial sample.**

*We removed "only" from the sentence.*

**P.7 L18. So far there was no mention about this modelling. Probably it should be mentioned somehow in introduction.**

*In the Introduction, we now added:*

*"The timing and intensity of Saharan dust depositions were simulated using two independent models (ALADIN-Climate and NMMB/BSC-Dust)."*

**P.7 L27 3.1 Modelled dust depositions?**

*We changed the title of this paragraph to: "Modelled dust deposition events"*

**P.7 L36-38 repetition.**

*We removed these two sentences.*

**P.8 L2 this strong dust event? (singular?)**

*We replaced "these" with "this".*

**P.9 L4 than P9. L6 Can you please explain why Crocus model shows a 100 mm SWE in Dec 2013 while this was not observed. Solid precipitation is one of the input parameters isn't it?**

*The GMON sensor was installed in 2013. During the first weeks, we had some problems with the power supply, so data were not recorded. In the caption of Figure 3, we added:*

*"SWE data are missing in December 2013 because of problems with the power supply."*

**P10. L.14 This is just one possible explanation though quite doubtful as particles are still quite large to be washed out that simple. It would've been great to see the description (photograph) of the snow pit in 2016 and to see how it corresponds with modelled structure. Additional samples collected from these dust layers separately could've helped. Another interesting question is how the local mineral particles (rocks, soil, vegetation. . .) affect snow melting. The mass can be substantial in the snow pack, but of course it will not be modelled by dust deposition model.**

*We agree with this comment. Unfortunately, we don't have high quality photographs of the snow pit. During the last two years, we've been visiting regularly the site and collecting multiple samples of snow containing dust.*

*In the manuscript, we added:*

*"This deeper layer can be probably due to the eventual scavenging of small dust particles by meltwater, or to other undetected processes".*

*The effect of larger particles on the snow melting is discussed in the answer to the following comment.*

**P.10 L18 The tail in distribution most likely is due to input from local particles. Looking at the photograph there are many rocks and vegetation around the site and Coulter Counter analysis do not distinguish between particles of different nature. This is quite an important issue. If the total mass concentration of mineral particles considered, than highest input would be from the small number of large particles.**

*We cannot exclude that large particles of local origin can be deposited on snow (we acknowledged it in pg 12 ln 16). Recently, we installed different deposimeters for evaluating the input of local and remote particles to the snowpack. In the future, Crocus model could be also modified to account for larger particles of local origin. In the text (pg 12 ln 16), we added:*

*"A contribution of large particles of local origin cannot be excluded, and it may have a strong influence on snow melting. At the moment, we don't have enough data to decouple the effect of large and small particles on snow albedo"*

**P.12 L4 Can you please clarify a bit more how exactly BC data were used. Was it an input to Crocus model? How large was the impact compare to dust. Isn't it the largest source of uncertainty? Can the BC signal be separated from the natural dust? Later in the text you mostly discuss the influence of the impurities without specifying.**

*Black carbon (soot) fluxes was one of the inputs of Crocus model (see Section 2.2). For decoupling the effect of dust and black carbon, Crocus can be run taking in account dust and black carbon separately (see Tuzet et al. 2017). In our simulations, both impurity fluxes are considered. Since we don't have direct measurements of black carbon in snow at our experimental site, we cannot exclude a possible influence on snowmelt. In the new version of the manuscript, we added:*

*"The role of black carbon in Alpine snow still represents a great uncertainty in snow modelling and climate prediction in the Alps. While the role of industrial black carbon on post-industrial glacier retreat has been debated (Painter et al. 2013; Sigl et al. 2018), its role on seasonal snow melting has not been studied in the European Alps."*

*References:*

*Painter, T. H., Flanner, M. G., Kaser, G., Marzeion, B., VanCuren, R. A., & Abdalati, W. (2013). End of the Little Ice Age in the Alps forced by industrial black carbon. Proceedings of the National Academy of Sciences of the United States of America, 110(38), 15216–21. https://doi.org/10.1073/pnas.1302570110*

*Sigl, M., Abram, N. J., Gabrieli, J., Jenk, T. M., Osmont, D., and Schwikowski, M.: 19th century glacier retreat in the Alps preceded the emergence of industrial black carbon deposition on high-alpine glaciers, The Cryosphere, 12, 3311-3331, https://doi.org/10.5194/tc-12-3311-2018, 2018.*

**P.12 L17 new paragraph? or maybe it's better to introduce a separate section on SDI**

*We added a new paragraph on SDI data and simulation.*

**P.14 L23 I doubt that it's a good argument to compare advancement in snow melt to distances from the deserts. You can either compare average (long-term) deposition rates or differences in snow duration reduction with similar dust concentrations.**

*We modified the sentence according to your comment. Now it reads:*

*"Despite the different deposition rates in the Alps, the advancement of the snowmelt owing to dust is comparable with published results regarding the Western US. This is true at least for one season (2015/2016), characterized by a major Saharan dust deposition."*

**P.14 L26 Is it possible to compare bulk concentrations (e.g. CC results) with the deposition modelling results?**

*A numerical comparison with Crocus prediction is provided in pg 13 ln 7. As showed in Tuzet et al. 2017, the concentration of impurities within the snowpack can be directly compared with Crocus predictions. In that case, dust concentrations were underestimated, while BC concentrations were overestimated. Our results show that observed dust concentrations were reasonably comparable with simulated ones. Also considering the large spatial mismatch between the point measurements and the ALADIN fluxes predictions.*

**P.14 L32-35 This is again partly repetition from the introduction. As well as in the next paragraph. Trends are not discussed in this paper at all so it can go to introduction.**

*We agree with this comment. But we prefer to lose this (and the following) sentence, since the introduction is already lengthy, we prefer not to add further text to it. We also removed the sentence in line 1-2 (pg 15), since it is repeated in the following paragraph.*

**P.15 L.8-10 This is a bit exaggerated. If the average over 82 years is late May than I believe snow disappeared in early May a number of times. Or maybe not? How many times exactly? So how rare such snow duration actually is? is it really extremely short? The next part of the paragraph is again an introduction-like and can be possibly moved up there too. The importance of the snow duration shifts is explained there. In results section it's better to discuss the exact results.**

*We used the expression "extremely short" because the first important snowfalls occurred in January for the 2015/2016 season. Considering this coupled with an earlier snowmelt due to dust depositions, this season is characterized by a snow cover duration of 4 months, over an average of 7 months. We acknowledge that the term is a little bit strong, so we replaced "extremely" with "very".*

**P 16 L.20-27 repetition of the introduction**

*We prefer to keep these introductory sentences in this chapter. They are important for putting into context the geochemical characterization of dust.*

**P.20 L. 9-11 I'd suggest to rephrase or remove this. This large topic needs much more regular analysis suitable methodology etc. It just sounds a little bit speculative.**

*We removed this sentence according to your comment.*

*Best regards,*

*Biagio Di Mauro and co-authors*

---

## Author Comment (AC3) · 5 Mar 2019

Reviewer: Dr. Marion Greilinger

Authors responses are in *italic*, Reviewer's comments are in **bold**. Line and figures numbers refer to the track-changes version of the manuscript.

**Review of:**

**Title: Saharan dust events in the European Alps: role on snowmelt and geochemical characterization**

**My recommendation**

**Major revisions due to general and specific comments listed below.**

**The authors investigate the input of mineral dust (MD) on the geochemistry as well as the impact on snowmelt in the Aosta Valley, Italy at 2160m a.s.l. within the accumulation periods 2013/14, 2014/15 and 2015/16. The study investigate the evolution of snow melt off via in-situ observations, digital images, AWS data and modelling. Besides the investigation of a snow darkening index representative for MD on the snow surface, a geochemical characterization from MD affected and non-affected snow was presented as well. Authors observed a shortening of the snow season, concluding that MD accelerate snow melt-out dates.**

**The addressed topic is of interest for the Cryosphere community, but also for the climate modelling (e.g. surface albedo feedback) and remote sensing community (e.g. validation and calibration of satellite images).**

*Dear Dr. Greilinger, thank you for the positive evaluation of the manuscript. We have carefully considered each of the Reviewer's comments and suggestions. The Reviewer will find below the responses to general and specific comments.*

**General comments:**

**My major concern is the structure and the "red line" throughout the manuscript. The manuscript suffers from many repetitions and the text does not account to the corresponding headline. Therefor it is very hard to read and needs a lot of scrolling to other passages to follow the "story" behind. It would be of much help for the reader and hence also of much more interest if this would be revised and shortened rigorously (therefore major revisions). Authors should think of splitting the results from the discussion into separate section. The manuscript might get ab bit more reader-friendly. Besides, citations seem to be sometimes randomly used whereas they are missing at points were there should be a quote. Authors should cite from recent to past or vice versa, but consistently throughout the manuscript. Details on the general comments raised above can be found in the specific comments below.**

*We removed repetitions along the manuscript, and we shortened it as suggested also by the other reviewer. Many paragraphs were moved in order to render the manuscript more fluid and reader-friendly. We carefully revised the citations in the manuscript. Regarding Section 3, we prefer to keep the results and discussion tied together in our paper.*

**Specific comments:**

**P1 L18 change "snowpack in a..." to "snow packs at a..."**

*We modified accordingly*

**P1 L28 ff Aren't these the results from the comparison of Crocus model results without impurities vs. observations? Otherwise to which reference do the values of 38 days etc. refer?**

*Yes, they are referred to the comparison between the snow depth from the model without impurities and the observed one. The sentence now reads:*

*"In our case study, the comparison between modeling results and observation showed that impurities deposited in snow anticipated the disappearance of snow up to 38 days for the 2015/2016 season that was characterized by a strong dust deposition event, out of a total 7 months of typical snow persistence"*

**P1 L34 Include also the importance on snow albedo feedback**

*We modified accordingly, the sentence now reads:*

*"We conclude that the effect of the Saharan dust is to anticipate the snow melt-out dates through the snow-albedo feedback. This process is known to have a series of further hydrological and phenological feedback effects, that should be characterized in future research"*

**P2 L7 remove sentence "These phenomena…"**

*We removed the sentence.*

**P2 L13 "…dust lowers THE snow albedo…"**

*We modified accordingly.*

**P2 L16 ff Which citation refers to which statement? One reference used twice in one sentence – maybe rewrite the sentence**

*We merged all the references at the end of the sentence, since they are all referring to the effect of dust on snow in Western US.*

**P2 L22 remove "s" from "centurys"**

*The spelling is already corrected.*

**P2 L30 maybe also include Greilinger et al here**

*We added this reference.*

**P2 L34ff remove "of the planet", change "Thanks to.." to "Due to.."**

*We prefer to keep the sentence as it is: "Even though the Alps are located at a distance of about 3000 km from the largest desert of the planet". In*

*the following sentence, we replaced "thanks to" with "due to" according to your comment.*

**P2 L40 "…precipitation and HENCE dust scavenging…"**

*We modified accordingly.*

**P3 L40 define LAPs here**

*We already defined the acronym LAPs in pg2 ln14. We added here which kind of LAPs were considered in the study (mineral dust and black carbon). The sentence now reads:*

*"[...] which can incorporate the effect of LAPs (mineral dust and black carbon) in snow [...]"*

**P4 L11ff "…was installed in 2009 measures air temperature (HMP45, Vaisala Inc.) and snow height (ultrasonic sensor SR50A, Campbell Scientific Inc.)."**

*We prefer to keep the sentences separated, since the AWS measures a variety of variables described below in the paragraph (not only air temperature and snow depth).*

**P4 L18 – P5 L2 belongs to introduction**

*We shortened the paragraph and we used it to introduce the use of digital images in the methodology section. The sentence now reads:*

*"In recent years digital images analysis was applied to the monitoring of vegetation phenology (Julitta et al., 2014; Migliavacca et al., 2011; Richardson et al., 2007), landslides, glaciers (Jung et al., 2010) and snow (Corripio, 2010; Dumont et al., 2011; Hinkler et al., 2010; Parajka et al., 2012). Regarding the two latter, using digital cameras researchers successfully retrieved snow albedo and snow cover in alpine areas."*

**P5 L3 Include new subsection 2.2 RGB images or digital images or similar**

*We introduced a new section:*

*"2.2 Digital images analysis"*

**P5 L6 rephrase "…and the same view scene was repeatedly captured" What would you like to say?**

*We meant that the camera is fixed, and the same scene is repeatedly photographed. The sentence now reads:*

*"and the same scene was repeatedly photographed"*

**P5 L7 ..."format WITH a resolution of…AND three-color channels (red, green, blue)…)**

*We modified accordingly.*

**P5 L9 Just as suggestion, it is always nice to refer to UTC. If you use local time, please specify time zone.**

*We used local time, that is in "UTC+1". We added this information in the manuscript:*

*"The images were collected from 10 am to 5 pm (local time: UTC+1), with an hourly temporal resolution."*

**P5 L16 "Following Di Mauro et al. (2015) and Ganey et al. (2017) SDI was correlated…distribution of deposited impurities  …"**

*Actually, in Di Mauro et al. 2015 we developed the regression models from field spectral data and radiative transfer modeling to link SDI and mineral dust concentration in snow. Then Di Mauro et al. 2017, and Ganey et al. 2017 used the index for mapping different kind of impurities from space. For these reasons, we prefer to keep the sentence in its original form.*

**P5 L17 What do you mean with "and from hypospectral imagery of ice cores"?**

*In Garzonio et al. 2018, we calculated the index from hyperspectral images acquired on an ice core drilled in the Alps for representing a time series of impurities deposition on a glacier. The high spectral resolution of these images allowed the calculation of different spectral indices.*

*For clarity, we replaced "imagery" with "images".*

**P5 L24 "using THE SURFEX…"**

*We modified accordingly.*

**P5 L25 "…estimation AS WELL AS numerical…"**

*We modified accordingly.*

**P5 L28 "…and mass transfer between the snowpack and the atmosphere as well as the snowpack and the ground…"**

*The model simulates also energy and mass transfer within the snowpack, the sentence now reads:*

*"Snow dynamics are represented as a function of energy and mass-transfer within the snowpack, between both the snowpack and the atmosphere, and the snowpack and the ground below"*

**P5 L31 "…snow properties, LAPs concentrations and…"**

*We modified accordingly.*

**P5 L33 "…and accounts for…and impurities such as dust and black carbon."**

*We modified accordingly.*

**P5 L34 "…TARTES was used to calculated SDI…" How was this done?**

*SDI was calculated using the formulation proposed in Di Mauro et al. 2015. The sentence now reads:*

*"Snow spectral albedo simulated with TARTES was used to calculate SDI (using the formulation proposed in Di Mauro et al. 2015), and it was compared with SDI calculated from the digital camera"*

**P6 L15 "…a few meters apart from the AWS."**

*We modified accordingly.*

**P6 L16 "…from a pit at depths of…"**

*The sentence now reads:*

*"For each snow pit, we collected a surface samples at 0 cm, and three samples at depths equal to 20, 40, and 60 cm from the surface"*

**P6 L18 remove "successive"**

*We modified accordingly.*

**P6 L20 "…particles between 2 and 60µm (equivalent spherical diameter)."**

*For clarity, we prefer to keep this sentence in its original form.*

**P6 L22 reference why you use 2.5G/cm³, Why exactly this number?**

*This is the common value used since the very early studies about the atmospheric mineral dust content extracted from ice cores (Hänel, 1968; Royer et al., 1983). It is slightly lower than the average continental crust density (that is about 2.9 g cm-3), since it was assessed that the mineral assemblage that characterize mineral aerosols is lighter than the average continental one (Hänel, 1968).*

*We deem that, in the manuscript, the reference to Ruth et al. 2008 is exhaustive.*

*–Hänel, 1968: The real part of the mean complex refractive index and the mean density of samples of atmospheric aerosol particles*

*–Royer et al., 1983: A 30000 year record of physical and optical properties of microparticles from an East Antarctic ice core and implications for paleoclimate reconstruction models*

*–Ruth, U., Barbante, C., Bigler, M., Delmonte, B., Fischer, H., Gabrielli, P., Gaspari, V., Kaufmann, P., Lambert, F., Maggi, V., Marino, F., Petit, J.-R., Udisti, R., Wagenbach, D., Wegner, A. and Wolff, E. W.: Proxies and Measurement Techniques for Mineral Dust in Antarctic Ice Cores, Environ. Sci. Technol., 42(15), 5675–5681, doi:10.1021/es703078z, 2008.*

**P6 L25-P7 L17 could be shortened, many passages not necessary. It is the Data and methods section!**

*As suggested also by the other Reviewer, these paragraphs were moved to Section 3.3*

**P7 L23 "…in 'strong' events with dust deposition fluxes…and 'weak' events with lower concentrations."**

*We modified accordingly.*

**P7 L36-P8 L2 removes paragraph, it is the Data and methods section!**

*We removed lines 36-38 pg. 7 as suggested also from the other Reviewer. We prefer to keep here the other sentence since it describes Figure 2, which is a Result.*

**P7 L10 remove first sentence**

*We removed the sentence accordingly.*

**P7 L11 "…variables observed at Torgnon station and simulated with the Crocus model using…"**

*We modified accordingly.*

**P8 L15 "In Figure 3d …"**

*We modified accordingly.*

**P8 L16ff Remove sentences "Strong and weak…" as well as "ALADIN CLiamte…"**

*We modified accordingly.*

**P8 L17 You found a good agreement between the qualitative information, but how about the quantitative?**

*For this preliminary comparison between NMMB/BSC-dust and ALADIN we were interested in the agreement of the timing of dust events. Results showed that both models predicted at least two strong events in the same periods (February 2014, and April 2016). Several weaker events were also detected by both models. A quantitative evaluation of this comparison is actually out of the scope of our manuscript.*

**P8 L27 Please be more explicit why results before explain the large different in snow melt out dates.**

*We now added this sentence to better explain the higher concentration simulated in surface snow:*

*"This can be due to the longer duration of the dust event in April 2016, and may also explain the large change (38 days) in the snow melt-out dates observed in the data"*

**P9 Figre3 I personally have difficulties to read and interpret Figure 3d and especially Figure 3e. Maybe explain in more detail in the text (and/or legend) what is the shaded area and what is the colored (reddish, yellowish) area?**

*We added further details in the legend of Figure 3, now it reads:*

*"Figure 3 a)-b)-c): time series of albedo, snow water equivalent (SWE), and snow depth (SD) measured with the AWS and simulated with Crocus model including and excluding the impact of LAPs. SWE data are missing in December 2013 because of problems with the power supply. d): dust fluxes simulated with ALADIN (maroon bars, note that the scale is inverted), and strong (large stars), and weak (small stars) dust events simulated with NMMB/BSC-Dust. e): dust concentration (µg/g) in the snowpack (yellow to black palette) simulated with Crocus and superimposed on the snow depth profile (grey shaded area). f): surface concentration (averaged over the first 10 cm) of dust simulated with Crocus."*

**P9 Table1 It would be also nice to show the correlation with the Crocus model without impurities**

*We added this information. We now created a Figure (i.e. Figure 4) for comparing $R^2$ and RMSE for Crocus simulation accounting and not accounting for the impact of LAPs. Here the new figure:*

[Figure]

*Figure 4 Comparison between snow depth (SD), snow water equivalent (SWE) and albedo observed from the AWS station in Torgnon and simulated with Crocus accounting and not accounting for the impact of LAPs on snow*

*Furthermore, we realized that that was an error in the time series of snow albedo in Figure 3 of the manuscript. We now corrected the series. Here the new Figure 3:*

[Figure]

*Figure 3 a)-b)-c): time series of albedo, snow water equivalent (SWE), and snow depth (SD) measured with the AWS and simulated with Crocus model including and excluding the impact of LAPs. SWE data are missing in December 2013 because of problems with the power supply. d): dust fluxes simulated with ALADIN (maroon bars, note that the scale is inverted), and strong (large stars), and weak (small stars) dust events simulated with NMMB/BSC-Dust. e): dust concentration (μg/g) in the snowpack (yellow to black palette) simulated with Crocus and superimposed on the snow depth profile (grey shaded area). f): surface concentration (averaged over the first 10 cm) of dust simulated with Crocus.*

**P9 L12 "...8.5μm for snow samples collected at 20cmand 40cm depth, instead…"**

*We modified accordingly.*

**P9 L15 Remove the sentence "At the bottom…"**

*We modified accordingly.*

**P9 L16 Authors say that results are comparable with other studies. Please give some numbers what others found, not only the citation.**

*We added the size distribution found by the referenced studies. Now the sentence reads:*

*"Dust size distributions are compatible with other measurements of dust enclosed snow and ice in the Alps (3-5 µm, Maggi et al., 2006), and in Caucasus (1.98-4.16 µm, Kutuzov et al., 2013). Differences between our samples and these studies may be ascribed to the different altitude of the samplings."*

**P9 L17 "Samples shown in…"**

*We modified accordingly.*

**P10L8 – P11L27 Please work through the whole section. Parts of the text are already mentioned before, conclusions drawn here are not obvious for the reader. Where exactly do I see the marked change in snowmelt rate and the induced earlier snowmelt in Figure 3e? Here you also mention already some conclusions. It is the Results and discussion section!**

*In these paragraphs, we present a focus on the 2015/2016 season. So, we briefly resume the role of dust on snow in this season. We removed some repetitions in the text. The marked change in snowmelt rate is clearly visible from the drop of snow depth in Figure 6. We removed the repetition in the following sentence. The sentence now reads:*

*"[..] a marked change in snowmelt rate is observed from the snow depth series around the 20th of April (Figure 6)."*

**P13 Figure5 "…data are also shown (black line)."**

*We modified accordingly.*

**P13 L10 "In the upper part of Figure 6 …"**

*We modified accordingly.*

**P14 L1 "In the lower part of …"**

*We modified accordingly.*

**P14 L5 Why are you sure that the red line is IN the pit? Couldn't this be also a shadowing effect of e.g. an uneven surface? Why should the February event be visible only in the area of the pit?**

*We just provided a possible interpretation of that pattern in the snow pit. The February event may be visible only in that area because it was then buried from new snow during the season. We modified the sentence, that now reads:*

*"This can be possibly associated with the precedent 'weak' depositions from February and March, which were concentrated in a thin snow layer by melting during early spring"*

**P14 L14 Which non-linear model? Explain and describe the model of Di Mauro et al. 2015 shortly.**

*We added further details on the nonlinear model that we used. Now the sentence reads:*

*"Using this information, we inverted the nonlinear (rational) model developed in Di Mauro et al. (2015) that links mineral dust concentration and SDI values, and we obtained an estimated dust concentration equal to 56 $\mu g_{dust}$ $g^{-1}_{snow}$."*

**P14 L24-L31 repetition and extensive discussion (maybe start a separate discussion section related to the sections in the results.**

We removed repetitions from this paragraph.

**P14 L32- P15 L2 Belongs to the introduction**

*We removed part of this text, since the issue was already present in the introduction section*

**P15 L3-L7 is an outlook, move to summary**

We believe that this outlook sentence is better suited for this section, because it is meant to put our results in the broader context of long-term monitoring of dust and black carbon depositions.

**P15 L8-L18 another discussion block**

*As we stated in the answer to your general comment, we prefer to keep the results and discussion tied together.*

**P15 L20-L26 move to introduction**

*We prefer to keep these introductory sentences in this chapter. They are important for putting into context the geochemical characterization of dust.*

**P15 L27 "The analysis of the elemental composition allowed…"**

*We modified accordingly.*

**P15 L28 Is the threshold of definition of major and minor components referring to > or < than 1% of the average crust composition set by the authors? Reference?**

*This definition is widely used in the geochemistry scientific community, and it can be found in any geochemistry textbook (e.g. Geochemistry, W.M. White, Wiley-Blackwell). The definition is already reported in the manuscript at pg 18 ln 10.*

**P16 Table2 state somewhere in the legend or in the plot that SH1 is the dust affected and SH2 the clean snow! This would help the reader. Otherwise readers have to go back to the Methods section to check this. What are the value in the brackets? Why are some elements given in % mass fraction and others in µg/g? This makes it difficult to compare.**

*In the caption of Table 2 (that became Table 1 in the revised version of the manuscript) we added the information on SH1 and SH2, and the meaning of values in the brackets (that are measurements uncertainties). Furthermore, we converted all elements concentration to µg/g. Now the caption of Table 1 reads:*

*"The elemental composition of SH1 (snow sample containing mineral dust) and SH2 (clean snow sample). Data are expressed in terms of µg g-1 and are referred to the mass of the extracted material, not to the considered snow volume. Values in brackets are measurement uncertainties. Normalized concentrations were calculated considering the Upper Continental Crust as a reference (Rudnick and Gao, 2003).*

*Regarding the description of errors, in the methods section we now added:*

*"For a complete description of the method, including the estimation of errors, see Baccolo et al. (2015, 2016)."*

**P17 L1 "Concentrations of major elements normalized to the upper continental crust composition are shown…"**

*We modified accordingly.*

**P17 L9 "…see in Figure 7c."**

*The comparison was made in each sub plot of Figure 8, not only Figure 8c.*

**P17 L13 For Fe this is even more than 30 times if I am not mistaken.**

*Actually, Fe is 222 times more concentrated in SH1 with respect to SH2 (40000 µg/g for SH1, versus 180 µg/g for SH2). For this reason, we wrote that the "absolute concentrations that are more than two orders of magnitude higher".*

**P17 L23 remove the sentence "This is related to…" This is discussed again few lines below**

*We modified accordingly.*

**P17 L27 remove "not with the first one"**

*We modified accordingly.*

**P17 L35 Actually it is not the Ca which is affecting the pH but the related Carbonate! Include here the Carbonate discussion from L23**

*We modified the sentence, that now reads:*

*"The Ca component of carbonates, beside affecting soil pH and improving soil structure, have important effects on ecosystem physiology (Schaffner et al., 2012)."*

**P17 L37 remove the bracket**

*We modified accordingly.*

**P17 L35-P18L3 maybe rephrase the whole paragraph, difficult to see what the authors like to say**

*We rephrased the whole paragraph, shortening it. Now it reads:*

*"For both elements, SH1 shows notably higher concentrations (see Table 1). This requires more attention and further studies to understand the feedback of Saharan dust deposition on biogeochemistry of high-altitude ecosystems"*

**P18 Figure7 What is the y-axis in Figure 7b? Remove the sentence "They are intended here…" from the legend. Remove "…, presenting anomalously high normalized concentrations;" Remove everything after "…listed following…" What is meant with "incompatible elements (with respect to Fe)? Indicate here also the nomenclature of SH1 and SH2 to be consistent throughout the manuscript.**

*We added the legend in the y-axis of Figure 8b, and we removed the text from the caption. Regarding the "incompatible elements" you can find further details in the referenced papers (Sun and McDonough, 1989). We decided to keep the text regarding the element listing, since it is not straightforward for non-specialists. We also added SH1, SH2, and Saharan sources (Moreno et al. 2006) in the legend of figure 8.*

*Here the new Figure 8:*

[Figure]

**P18 L16 include sentence "They are…" already in the first sentence of the paragraph in line 13.**

*We modified accordingly.*

**P18 L23 Remove the sentence "Given the position…"**

*We modified accordingly.*

**P19 L1-L10 repetition to earlier passages**

*We shortened the paragraph, removing the repetition regarding anthropogenic activities in the Po valley.*

**P19 L12 What does this "incompatibility degree with respect to Fe" reveal? Why use this measure?**

*In Moreno et al 2006, this metric is used to characterize Saharan dust sources. In the manuscript, we already stated that this is useful understand the provenance and the geochemical signature of rock samples*

**P19 L13 Remove the sentences "As in the case..." until "low normalized concentrations." Repetition!**

*We modified accordingly.*

**P19 L33 The content of the next section is not a conclusion but a summary! Please stet the conclusion you draw based on your work more explicitly.**

*In Section 4 (Conclusion) we included a summary of the findings of our paper in which the conclusions are already clearly stated. In this section, we also provide some future perspectives in the growing body of research focusing on the role of impurities on snow.*

**P19 L38 "…11days, respectively."**

*We modified accordingly.*

**P20 L3 See also http://www.aaqr.org/article/detail/AAQR-18-03-ACPM-0116 to confirm the Sahara dust event. Include citation.**

*We included this citation.*

**P20 L9 But the fingerprint of the local sources plays also a role. Please state this here in the text.**

*The sentence now reads:*

*"These results demonstrate that through an accurate geochemical characterization of dust deposited on the Alps, it is possible to identify the different Saharan sources involved in the single transport events, but the fingerprint of the local sources may play also an important role"*

**P20 L20 Maybe you find something in here https://onlinelibrary.wiley.com/doi/abs/10.1034/j.1600-0889.49.issue1.4.x**

*Thanks for the suggestion, we will take this into account for future analysis on the geochemistry of snow in the Alps.*

*Best regards,*

*Biagio Di Mauro and co-authors*

---

## Editor Decision (ED1)

Dear authors

I am pleased with your replies to the reviewer's comments. In agreement with the reviewers I still see room for more concise presentation of your research. Moreover, the manuscript would definitely profit from some rigorous editing. Nevertheless, I am prepared to accept the manuscript after some technical corrections. Below you will find some mainly editorial comments (not exhaustive).

Page 1, line 18: "elevation" is the preferred term (several time elsewhere in the manuscript)

Page 1, line 30: "snow duration" or "snow season" is the preferred term (several time elsewhere in the manuscript)

Page 1, line 29: I suggest replacing "anticipating" (several time elsewhere in the manuscript)

Page 1, line 31: I suggest rewording: ... the snow melt-out was 18 and 11 days earlier, respectively.

Page 1, line 32: I suggest rewording: … is expected to reduce snow cover duration

Page 2, line 16: snowmelt, "one month of earlier snowmelt",

Page 3, line 14: … most dust depositions occur by wet deposition (mainly snowfalls)…

Page 3, line 20: "snowfalls" (several time elsewhere in the manuscript)

Page 3, line 26: "snowmelt" (several time elsewhere in the manuscript)

Page 3, line 28: "accelerated snowmelt due to dust …" (several time elsewhere in the manuscript)

Page 4, line 1: the study site ... at an elevation…

Page 4, line 21: Unclear what you refer to, suggest rewording.

Page 5, line 5: during the hydrological years 2013-2016 (several time elsewhere in the manuscript)

Page 5, line 20: "snowpack"

Page 5, line 37: "snowfall"

Page 9, line 18: …buried by subsequent snowfalls.

Page 9, line 20: …is resurfacing towards the end of the season, …

Page 10, Figure 3 and 4: Labels (a,b,c,…) in figures are missing.

Page 10, Figure 3: I suggest changing units for dust flux (to replace $10^{-8}$).

**Page 10, Figure 4c: Can you please explain why the RMSE is smaller for the Crocus simulations without considering LAPs than for the simulations considering LAPs.**

Page 10, lines 13-14: Suggest rewording.

Page 11, line 1: local larger particles?

Page 12, line 3: point measurement

Page 12, line 30: I suggest replacing "snowfields" (and elsewhere in the manuscript)

Page 14, line 14: measured continuously

Page 15, line 37: shown

Page 19, line 2: I suggest replacing "anthropic" by "anthropogenic" (and elsewhere in the manuscript)

Page 19, line 6: Unlcear, suggest rewording.

Page 19, line 27: … on snowmelt at a high-elevation site…

Page 19, line 28: Unclear, suggest rewording.

19 March 2019
Jürg Schweizer

---

## Author Response (AR2)

**Dear authors**

**I am pleased with your replies to the reviewer's comments. In agreement with the reviewers I still see room for more concise presentation of your research. Moreover, the manuscript would definitely profit from some rigorous editing. Nevertheless, I am prepared to accept the manuscript after some technical corrections. Below you will find some mainly editorial comments (not exhaustive).**

*Dear Editor,*

*thank you for the evaluation of our manuscript. Hereafter you find a point-by-point answer to your comments.*

**Page 1, line 18: "elevation" is the preferred term (several time elsewhere in the manuscript)**

*We replaced "altitude" with "elevation" in the manuscript*

**Page 1, line 30: "snow duration" or "snow season" is the preferred term (several time elsewhere in the manuscript)**

*We modified accordingly.*

**Page 1, line 29: I suggest replacing "anticipating" (several time elsewhere in the manuscript)**

*We modified accordingly.*

**Page 1, line 31: I suggest rewording: ... the snow melt-out was 18 and 11 days earlier, respectively.**

*We modified accordingly.*

**Page 1, line 32: I suggest rewording: … is expected to reduce snow cover duration**

*We modified accordingly.*

**Page 2, line 16: snowmelt, "one month of earlier snowmelt",**

*We modified accordingly.*

**Page 3, line 14: … most dust depositions occur by wet deposition (mainly snowfalls)…**

*We modified accordingly.*

**Page 3, line 20: "snowfalls" (several time elsewhere in the manuscript)**

*We modified accordingly.*

**Page 3, line 26: "snowmelt" (several time elsewhere in the manuscript)**

*We modified accordingly.*

**Page 3, line 28: "accelerated snowmelt due to dust …" (several time elsewhere in the manuscript)**

*We modified accordingly.*

**Page 4, line 1: the study site ... at an elevation**…

*We modified accordingly.*

**Page 4, line 21: Unclear what you refer to, suggest rewording.**

*The sentence now reads:*

*"Regarding the two latter, snow albedo and snow cover were successfully estimated using digital cameras in alpine areas."*

**Page 5, line 5: during the hydrological years 2013-2016 (several time elsewhere in the manuscript)**

*We modified accordingly.*

**Page 5, line 20: "snowpack"**

*We modified accordingly.*

**Page 5, line 37: "snowfall"**

*We modified accordingly.*

**Page 9, line 18: …buried by subsequent snowfalls.**

*We modified accordingly.*

**Page 9, line 20: …is resurfacing towards the end of the season, …**

*We modified accordingly.*

**Page 10, Figure 3 and 4: Labels (a,b,c,…) in figures are missing.**

*We added labels to these figures*

**Page 10, Figure 3: I suggest changing units for dust flux (to replace 10-8).**

*We now expressed ALADIN dust fluxes in µg/(m$^2$ s)*

**Page 10, Figure 4c: Can you please explain why the RMSE is smaller for the Crocus simulations without considering LAPs than for the simulations considering LAPs.**

*In the manuscript we added:*

*"For snow depth (Fig. 4a) and SWE (Fig. 4b), Crocus simulations with LAPs generally resulted in lower RMSE and higher $R^2$ with respect to Crocus simulations without LAPs. Instead, for snow albedo (Fig. 4c) RMSE resulted smaller for Crocus simulations without LAPs. We underline that these RMSE values are associated with very low explained variance ($R^2$~0.2 for the seasons 2014/15, 2015/16, and all years; and $R^2$=0.43 for the season 2013/14). Thus, Crocus simulations with LAPs perform better than Crocus simulations without LAPs in modelling snow dynamics at Torgnon."*

**Page 10, lines 13-14: Suggest rewording.**

*We split the sentence. Now it reads:*

*"Instead, dust particles found in snow at the bottom of the snowpack (60 cm depth) feature a mode of 3.2 μm"*

**Page 11, line 1: local larger particles?**

*Now the sentence reads:*

*" […] local particles with larger diameter"*

**Page 12, line 3: point measurement**

*We modified accordingly.*

**Page 12, line 30: I suggest replacing "snowfields" (and elsewhere in the manuscript)**

*We modified accordingly.*

**Page 14, line 14: measured continuously**

*We modified accordingly.*

**Page 15, line 37: shown**

*We modified accordingly.*

**Page 19, line 2: I suggest replacing "anthropic" by "anthropogenic" (and elsewhere in the manuscript)**

*We modified accordingly.*

**Page 19, line 6: Unlcear, suggest rewording.**

*The sentence now reads:*

*"The composition of a suite of elements found in trace is presented in Figure 8c"*

**Page 19, line 27: … on snowmelt at a high-elevation site…**

*We modified accordingly.*

**Page 19, line 28: Unclear, suggest rewording.**

*The sentence now reads:*

[revised manuscript text omitted]